# Tabular Data in Interactive and Conversational AI: A Survey of Foundations, Benchmarks, Systems, and Open Problems

## Abstract

Tabular and structured data underlie much of modern analytical work, yet natural language systems for interacting with such data have largely been studied in fragmented subfields. This survey studies that landscape under the broader problem of *conversational AI over tabular and structured data*: systems that support multi-turn, context-dependent interaction with tables, databases, spreadsheets, and hybrid table–text documents. We first clarify the problem setting by defining tabular data, conversational interaction, and the primary interaction modes that distinguish querying, translating, manipulating, and orchestrating over structured data, while treating exploration as a recurrent interaction pattern rather than a separate category. Using an explicit corpus-construction and evidence policy, we organize 106 unique cited works into five categories: *Foundations*, *Conversational Table Question Answering* (CTabQA), *Conversational Text-to-SQL* (CText2SQL), *Interactive Table Manipulation*, and *Agentic Table Systems*. Across these categories, we compare benchmark datasets, modelling paradigms, and evaluation practices, while tracing how closely related problems have often been studied under different task names, benchmarks, and research communities. Our synthesis shows recurring fragmentation in terminology, benchmark conventions, and modelling assumptions across the surveyed literatures, but we treat that fragmentation as a qualitative finding of the review rather than as a formal bibliometric result. Among the five categories, CText2SQL has the most standardized evaluation setup, judged by shared benchmarks, hidden test splits, and active leaderboards. Manipulation and agentic systems target tasks that resemble applied data work, such as editing spreadsheets and running multi-step analyses, but they are harder to evaluate, and we are not aware of user studies that confirm how well they match real workflows. Beyond category-specific findings, we identify three cross-cutting themes shared across the field: intent disambiguation and clarification, dialogue context tracking, and evaluation. These reveal a central mismatch between current benchmarks and realistic use: most systems are still optimized for short, clean, single-table interactions rather than long-horizon, ambiguous, multi-source analytical workflows. We conclude by synthesizing the field's main open problems, including unified evaluation, long-dialogue robustness, proactive clarification, interpretability, privacy, domain adaptation, and multi-table reasoning, and argue that progress will depend on moving from narrow task benchmarks toward integrated, user-centered conversational data systems.

## 1 Introduction

Tabular data, meaning information organized into rows and columns, is one of the most pervasive formats for storing and communicating structured knowledge. Relational databases (Codd, 1970) hold the operational records of governments, hospitals, and enterprises. Spreadsheets have been a primary analytical tool for millions of knowledge workers for decades (Scaffidi et al., 2005). Scientific papers report experimental measurements in tables. Financial disclosures present earnings, revenues, and risk factors in hybrid documents that interleave prose with numerical tables. Enterprise structured data, including relational databases and spreadsheets, accounts for a substantial share of organisational information infrastructure, yet natural

language processing (NLP), the scientific study of how computers understand and generate human language, has historically treated *text* as the primary medium of knowledge.

This gap is narrowing. Over the past decade, a growing body of research has asked a deceptively simple question: can a user interact with a table using plain language? Early work framed this as a single-turn *lookup*, where given a question and a table, the task was to find the answer (Pasupat & Liang, 2015). Later work extended this to *translation*, which involves converting a natural language question into SQL (Structured Query Language, a standard language for retrieving data from relational databases) so that answers could be retrieved programmatically (Yu et al., 2018). Both paradigms have produced rapid benchmark progress, with modern systems posting strong results on established evaluations (Gao et al., 2024; Pourreza & Rafiei, 2023).

Yet a fundamental mismatch remains between how these systems are built and how people actually work with data. Real interaction with tabular data is rarely a single question followed by a final answer. A financial analyst studying quarterly earnings does not ask one question and walk away; they probe, follow up, compare across rows, ask for clarification when a figure seems anomalous, request a reformulation when the first answer is ambiguous, and gradually build an understanding of the data through a sequence of exchanges. A physician reviewing a patient's lab results over time asks questions whose meaning depends entirely on what was asked before. A student exploring a dataset for the first time may not even know what question to ask until they have seen a few answers. This kind of iterative, goal directed, multiturn interaction is fundamentally conversational in nature, and it is precisely what existing systems are not designed to support.

## 1.1 The Conversational Gap

By *conversational* we mean interaction in which the meaning of the current input depends on prior turns, and in which the system may itself ask for clarification or propose a reformulation rather than only return an answer. Section 2.1 gives the full definition. The conversational dimension introduces challenges that single-turn systems are not equipped to handle. The system must maintain a *dialogue context*, which is a running representation of what has been said, what has been retrieved or computed, and what the user appears to be trying to accomplish. It must resolve *coreferences*, which are expressions in the current turn that refer to entities, values, or table cells mentioned in a previous turn. It must decide, in real time, whether the user's current question is sufficiently clear to answer or whether asking a clarification question would serve the user better. It must accumulate *numerical context* across turns, since a conversational financial analysis may require an intermediate result computed in turn three to answer the question posed in turn seven. None of these are problems that a single-turn question answering (QA) or Text-to-SQL system faces. Compounding these challenges, real tables are not clean. They may span multiple pages, contain merged cells and nested headers, mix numerical and categorical entries, embed footnotes, or appear alongside prose that qualifies their contents. A system capable of conversational interaction with such data must handle both the structural complexity of the table and the contextual complexity of the ongoing dialogue simultaneously.

## 1.2 Related but Fragmented Research Lineages

Research on these problems has not been idle. Over the past decade, at least five partially overlapping research lineages have been developing systems that address pieces of the conversational table AI problem, though none has framed the full problem explicitly in those terms. The *conversational table question answering* community has produced datasets such as SQA (Iyyer et al., 2017), which decomposes complex questions about Wikipedia tables into sequences of simpler follow-up questions; HybriDialogue (Nakamura et al., 2022), which grounds multiturn conversations in both tables and the accompanying text passages; PACIFIC (Deng et al., 2022), which introduces proactive clarification question generation as a component of financial conversational QA; and ConvFinQA (Chen et al., 2022), which requires chained numerical reasoning across the turns of a financial dialogue. The *conversational Text-to-SQL* community has produced SParC (Yu et al., 2019b) and CoSQL (Yu et al., 2019a), which decompose complex database queries into multiturn dialogues between users and SQL-generating systems, and a rich line of models for tracking SQL context across turns (Cai & Wan, 2020; Zheng et al., 2022; Qi et al., 2022). The *interactive spreadsheet*

*systems* community, operating largely within human-computer interaction (HCI), the study of how people interact with computational systems, has built tools such as SheetCopilot (Li et al., 2023a) and SheetAgent (Chen et al., 2025b) that allow users to modify spreadsheets through natural language instructions executed as sequences of software actions. The *natural language interfaces to databases* (NLIDBs) community has produced interactive visual analytics tools such as Eviza (Setlur et al., 2016) and NL4DV (Narechania et al., 2021) that allow users to explore data visually through conversational language queries. Most recently, the *agentic systems* community has developed systems built around large language models (LLMs), which are neural networks trained on large amounts of text that can generate, reason over, and act on language, that autonomously plan and execute sequences of database operations, spreadsheet edits, and data analysis steps in response to open-ended user goals (Zhang et al., 2023b; Tian et al., 2026).

In the taxonomy used in this survey, these lineages condense into four primary task communities plus one enabling layer: the spreadsheet-manipulation and NLIDB/visual-analytics lines are grouped together under *Interactive Table Manipulation*, while *Foundations* is treated separately as the methodological substrate beneath all four. Across the retained corpus, these lineages often frame closely related interaction problems with different terminology, benchmark conventions, and task boundaries. We therefore use *parallel but disconnected* as a careful qualitative characterization of the literature map assembled in this survey: the phrase captures recurring fragmentation in problem framing, benchmark design, and evaluation practice, not a standalone bibliometric claim. A dedicated citation-network study over the surveyed corpus would be a valuable follow-up project, but the present survey does not claim to have established that fragmentation quantitatively. The distinctive contribution of this survey is therefore not a blanket claim to be the first survey on tables, NL interfaces, or Text-to-SQL in general. It is narrower and more specific: we make multiturn, context-dependent interaction the organizing lens, and we use that lens to read across CTabQA, CText2SQL, interactive manipulation, and agentic table systems while treating Foundations as the shared substrate beneath them.

## 1.3 What This Survey Contributes

We make the following contributions:

**A unified taxonomy.** We organize the literature into five categories: (1) *Foundations*, covering methods for representing and encoding tabular data; (2) *Conversational Table QA* (CTabQA), covering multiturn question answering grounded in tables or mixed table-text documents; (3) *Conversational Text-to-SQL* (CText2SQL), covering context-dependent database query generation across dialogue turns; (4) *Interactive Table Manipulation*, covering systems that allow users to modify, create, and explore tables through dialogue; and (5) *Agentic Table Systems*, covering autonomous LLM-based agents that plan and execute multistep operations over structured data.

**A comprehensive literature map.** We survey 106 unique cited works spanning these five categories, tracing the evolution of each from early dataset construction papers through the latest LLM-based systems. We draw explicit conceptual connections across communities that have often been discussed separately and that still lack shared benchmark, terminology, and evaluation standards.

**A structured benchmark analysis.** We compare datasets across the five categories along a common set of dimensions, including task type, number of dialogue turns, domain, data size, table source, and evaluation metric, in Table 2. This analysis makes visible both the coverage and the gaps in existing evaluation infrastructure. Within the corpus we surveyed, we are not aware of any benchmark that spans all five categories.

**A cross-cutting analysis of shared challenges.** We identify three challenges that arise in every category and examine how each community has addressed them: (i) intent disambiguation and clarification, meaning the problem of determining what the user wants when their request is ambiguous; (ii) dialogue context tracking, meaning the problem of maintaining and updating a representation of what has been discussed across turns; and (iii) evaluation, meaning the challenge of measuring progress when task types, data formats, and interaction modes differ substantially across sub-areas.

**An agenda for future research.** We close with six concrete open problems, grounded in specific gaps identified throughout the survey, that we see as promising targets for the next generation of work in this area.

## 1.4 Review Protocol and Evidence Policy

This survey is a structured narrative literature review, not a systematic review or meta-analysis. We therefore do not claim PRISMA compliance or provide a PRISMA screening flow. However, following the transparency aims of PRISMA 2020 (Page et al., 2021), we report the corpus-construction procedure, query families, inclusion and exclusion rules, search-yield checks for reproducible indices, citation-correction record, deduplication record, and full audited corpus in Appendix B. We assembled the corpus in two ways: keyword search across ACL Anthology, arXiv, Semantic Scholar, OpenReview, and project pages, combined with backward and forward citation searching from the category-defining works. Most works were reached through citation searching rather than by the keyword queries alone, so we do not claim a systematic database-only search. We froze the corpus on March 27, 2026, and retained a paper if it contributed to multiturn or interactive work over structured data or gave background needed for one of the five categories. We excluded knowledge graphs, unstructured document dialogue without table grounding, and single-turn tasks except as predecessors or background. Preprints are included when they introduce benchmarks, systems, or public results, and the text labels their status. The survey cites 106 unique works, verified after deduplication; Appendix B lists all of them so that every citation key can be checked against its source.

## 1.5 Scope and Exclusions

The survey's scope is bounded by two criteria. First, the data must be *structured*, meaning organized into rows and columns, whether in a relational database, a spreadsheet, a financial table embedded in a PDF document, or a result table in a scientific paper. We do not cover knowledge graphs, which are graph-structured databases that represent entities and relationships as nodes and edges, as they pose related but distinct challenges and have their own extensive survey literature (Ji et al., 2022). Second, the interaction must be *multiturn or interactive*, involving at minimum the possibility of context-dependent interpretation, coreference resolution, or system-initiated communication beyond a single question and answer exchange. We treat single-turn table QA (Pasupat & Liang, 2015; Nan et al., 2022) and single-turn Text-to-SQL (Yu et al., 2018) as background and motivation rather than primary subject matter. We exclude table *generation* tasks, where systems synthesize table content from scratch (Fang et al., 2024), and table-based *fact verification* (Chen et al., 2020a), both of which are single-turn in nature and covered by prior surveys. Text-only conversational QA systems such as CoQA (Reddy et al., 2019) and QuAC (Choi et al., 2018) are discussed as structural analogues that helped establish the problem of multiturn QA, but they do not involve tabular grounding and are therefore not primary subject matter.

## 1.6 Relationship to Prior Surveys

Several recent surveys address adjacent territory but organize it differently: LLMs on tabular data broadly (Fang et al., 2024), NL-to-SQL methods and lifecycle (Liu et al., 2024), natural language interfaces for querying and visualization (Zhang et al., 2024b), deep-learning Text-to-SQL (Katsogiannis-Meimarakis & Koutrika, 2023), and conversational QA as a dialogue problem (Zaib et al., 2021). Table 8 makes the positioning explicit. Relative to these, our distinctive value is not covering every table-related task but centering multiturn, context-dependent interaction over structured data and using that lens to connect querying, SQL generation, manipulation, and orchestration.

This framing leads to a more modest but better-supported claim. Rather than asserting a generic "first to unify," we argue that the paper's contribution is an interaction-centered synthesis across adjacent literatures that are usually surveyed separately.

## 2 Background and Definitions

This section defines the two concepts the rest of the survey depends on: what we mean by *conversational* interaction (Section 2.1) and the typology of interaction modes that underlies the taxonomy (Section 2.2). Background on what counts as a table and on how tabular data differs from text is in Appendix C; a glossary of recurring terms is in Appendix A.

### 2.1 What Is Conversational Interaction?

We call an interaction *conversational* if it has at least one of four properties. **Context dependence**: the current input cannot be understood without prior turns, as in "and the year before?"; this is the property that separates conversational from single-turn table AI. **Coreference**: an expression refers to an entity, value, or cell from a prior turn, as when "what about its operating margin?" points to a company named earlier. **System-initiated communication**: the system acts beyond answering, for example by asking a clarification question or proposing an interpretation, which is *mixed-initiative* dialogue (Allen et al., 1999). **Iterative refinement**: the user adjusts a request across turns toward one evolving goal rather than issuing independent queries. Not every system shows all four: early dataset papers focus on context dependence and coreference, while recent agentic systems add refinement and initiative. We use these four properties to characterize systems consistently.

### 2.2 Interaction Types: A Typology

The taxonomy introduced in Section 3 is organised around four primary modes of interaction between a user and a table-grounded AI system, together with one additional pattern that cuts across them: exploration. Distinguishing these modes before the taxonomy section is important because the differences are not merely technical: they reflect different user goals, different system capabilities, and different notions of what constitutes a correct response.

**Querying.** The user asks a question and the system returns an answer extracted or computed from the table. The answer may be a single value, a set of values, or a natural language sentence. The user's goal is information retrieval, that is, to learn something from the table that was not already known. Conversational Table QA systems (Section 4) are primarily querying systems.

**Translating.** The user expresses an information need in natural language and the system translates it into a formal query language, most commonly SQL, which is then executed against a database to produce results. The key difference from querying is that the system's primary output is a program rather than a direct answer. Conversational Text-to-SQL systems (Section 5) are translating systems. The conversational challenge is generating a query whose meaning depends on the history of prior queries in the session.

**Manipulating.** The user instructs the system to change the contents or structure of the table, for example by inserting rows, deleting entries, reformatting columns, applying formulas, or reorganizing the layout. The user's goal is table modification rather than information retrieval, and a correct response is a modified table that faithfully reflects the user's instruction. Interactive table manipulation systems (Section 6) are manipulation systems.

**Exploration as a recurrent interaction pattern.** The user does not have a single sharply specified question but wishes to understand the structure, patterns, or anomalies in the data. The system may proactively suggest summaries, highlight unexpected values, propose visualizations, or generate descriptive narratives. This behaviour is the most open ended and the least standardised within the NLP literature. It is most visible in interactive visualization tools such as Eviza (Setlur et al., 2016) and NL4DV (Narechania et al., 2021). In the taxonomy of Section 3, we therefore treat exploration not as a standalone primary category but as a recurrent interaction pattern that most often appears within manipulation-oriented systems and, in practice, can surface in other categories as well.

**Orchestrating.** The user specifies a high-level goal and the system decomposes it into a sequence of sub-tasks, executes them autonomously using available tools, and returns a consolidated result. The user's input may be a single sentence that expands into dozens of discrete operations involving multiple tables, code execution, and intermediate reasoning. Agentic table systems (Section 7) are orchestrating systems.

These interaction modes are not mutually exclusive in practice. A real session might begin with exploration, move to targeted querying, include a table edit, and conclude with a compiled report. The taxonomy separates the four primary modes because different research communities have focused on different system outputs, and understanding which mode a given system addresses is essential for interpreting its benchmark results and identifying what it cannot do.

## 3  A Taxonomy of Conversational AI over Tabular Data

The literature on conversational interaction with tabular and structured data has grown across at least five research communities that have often been studied in partial isolation from one another. Each community developed its own terminology, benchmarks, and evaluation criteria for what is, at a deeper level, a shared problem: enabling users to interact with structured data through natural language over multiple exchanges. Unifying these communities requires a taxonomy that is principled enough to reveal their connections yet specific enough to respect the genuine differences in what each community has built.

We organize the literature into five categories. The first category, *Foundations*, covers methods for representing tables in a form that language models can process; it is the substrate on which the other four categories build. The remaining four categories correspond to the four primary modes of interaction introduced in Section 2.2: querying (CTabQA), translating (CText2SQL), manipulating (Interactive Table Manipulation), and orchestrating (Agentic Table Systems). Exploration is retained in the background typology as a recurrent interaction pattern, but not elevated here to a fifth primary task community. The reason is empirical rather than terminological: exploratory systems such as Eviza and NL4DV are most naturally studied as part of manipulation-oriented and visual-analytics workflows, not as a mature standalone research community with its own benchmarks and modelling lineage. We therefore track exploration as a design dimension that can appear within manipulation systems and, in practice, across other categories as well.

Three cross-cutting themes arise in every category and are treated separately in Section 8: intent disambiguation and clarification, dialogue context tracking, and evaluation. Figure 1 provides a visual overview of the taxonomy. Table 9 gives a compact characterization of each category.

### 3.1  A Running Example

One request can touch all five categories. Take: "Which product line had the sharpest margin decline after Q2, flag every quarter below 15% margin in the spreadsheet, and prepare a short note for the CFO." *Foundations* fixes how the workbook is encoded so the model can read its structure. *CTabQA* answers the margin-decline question, possibly over follow-ups like "only for Europe." *CText2SQL* does the same when the data lives in a database, translating each turn into context-dependent SQL. *Interactive Table Manipulation* makes the change, inserting the flag column and formulas. *Agentic Table Systems* orchestrate the whole task end to end, including deciding when to ask for clarification and drafting the note. The categories are complementary layers of one problem, not competitors.

### 3.2  Category 1: Foundations

The Foundations category covers methods that enable language models to process tabular data at all. These are not conversational systems in themselves; rather, they provide the representational substrate on which conversational systems are built. We include them as a distinct category because advances in table representation have consistently unlocked new capabilities in the categories above them.

The central challenge addressed by this category is that language models are trained on sequences of tokens, meaning words, subwords, and characters arranged in a linear order. Tables are not linear. A cell's meaning depends on its column and row position simultaneously, and a model that reads a serialized table as if it were

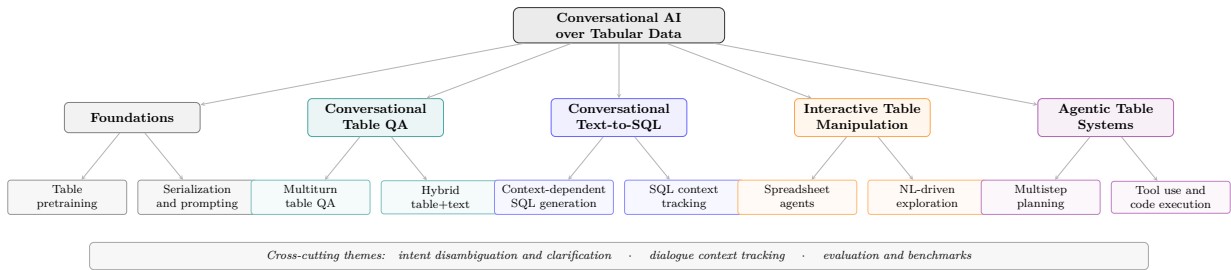

Figure 1: Taxonomy of conversational AI over tabular and structured data. Five categories are arranged below the root node, each with two representative sub-areas. CTabQA = Conversational Table Question Answering; CText2SQL = Conversational Text-to-SQL. The italicised bar at the bottom represents three cross-cutting themes that arise in every category, analysed together in Section 8.

running text may fail to recover this two-dimensional structure. Two broad lines of work have addressed this problem.

**Table pretraining** adapts language model pretraining to make models sensitive to tabular structure. TaPas (Herzig et al., 2020) extends BERT (Bidirectional Encoder Representations from Transformers, a widely used pretrained language model (Devlin et al., 2019)) with additional token-type (segment) embeddings that encode each token's row index and column index within the table, allowing it to answer table questions through weak supervision, meaning training using automatically derived rather than manually annotated labels. TaBERT (Yin et al., 2020) introduces vertical self-attention, a mechanism that connects tokens occupying the same column position across different rows. TAPEX (Liu et al., 2022b) takes a distinct approach, pretraining a model to function as a neural SQL executor by training it to produce the output of SQL queries over tables. OmniTab (Jiang et al., 2022) combines natural table–text pairs with synthetic question answering supervision, while UnifiedSKG (Xie et al., 2022) broadens the picture by framing 21 structured knowledge grounding tasks in a unified text-to-text setting rather than as isolated problems. More recent large-scale models such as TableGPT2 (Su et al., 2024) train dedicated table encoders jointly with a large language model, combining structural table understanding with the broad language capabilities of modern LLMs.

**Serialization and prompting** addresses how to present a table to a language model at inference time, which is the stage where a trained model processes new inputs rather than updating its weights. Approaches range from simple linearization, in which row contents are concatenated as a text string, to structured prompting strategies such as Chain-of-Table (Wang et al., 2024), which interleaves table operations with the model's intermediate reasoning steps, and Binder (Cheng et al., 2023), which connects the model to a symbolic execution environment so that arithmetic is handled by a code interpreter rather than the model itself. TableRAG (Chen et al., 2024) introduces a retrieval-augmented approach that allows models to selectively access relevant portions of very large tables rather than encoding the entire table in a single input, addressing the scale challenge of large tables. Table 10 provides a structured comparison of the key Foundations methods covered in this category.

The specific methods from this category are discussed inline as they arise in Sections 4 through 7, where each is introduced in the context of the conversational system that builds on it. The enabling-substrate status of Foundations is defined once in Table 9 and is not repeated in later tables.

### 3.3 The Four Interaction Categories

The other four categories correspond to the four interaction modes, and each is surveyed in its own section, so we only state the distinctive premise of each here. *Conversational Table QA* (CTabQA, Section 4) answers multiturn questions grounded in tables or hybrid table-text documents; it is the category closest to the survey's central problem. *Conversational Text-to-SQL* (CText2SQL, Section 5) translates each turn into SQL that depends on the SQL of prior turns, and has the most standardized benchmarks of the five. *Interactive Table Manipulation* (Section 6) changes the table itself through natural language, spanning

Table 1: The four interaction categories read along four capability axes. Each cell describes the typical case in that category, based on the systems and benchmarks surveyed here, and not every system in a category behaves the same way. Foundations is omitted because it is the enabling layer, not an interaction mode. These axes, rather than the community labels, are what determine which problems a system actually faces; Section 8 follows them in depth.

| Category (mode) | Clarification | Table state | Numerical chaining | Initiative |
|---|---|---|---|---|
| CTabQA (querying) | Modelled in PACIFIC, rare otherwise | Read-only | Central (ConvFinQA, PACIFIC) | User; system only to clarify |
| CText2SQL (translating) | Present in CoSQL, not tied to SQL state | Read-only | Low | User; system only to clarify |
| Manipulation (manipulating) | Rare; defaults used for underspecification | Mutable | Incidental (formulas) | User |
| Agentic (orchestrating) | Mostly autonomous; Harmonia is the exception | Mutable plus tool state | Via code execution or external memory | System |

spreadsheet agents and visualization interfaces. *Agentic Table Systems* (Section 7) take a high-level goal and plan and execute the steps autonomously, subsuming the other modes within one session. The rest of this section draws the connections among them.

### 3.4 Connections Across Categories

The five categories are not isolated silos. The clearest way to see what connects and separates them is to read them along four capability axes that cut across the community labels: whether the system can *ask* when intent is unclear (clarification), whether the table *state* is read-only or mutable, whether intermediate *numerical* results must be carried across turns, and who holds the *initiative* in deciding the next step. Table 1 maps the four interaction categories onto these axes.

Read this way, the field splits along lines the category labels hide. Clarification is studied in only a few places, so the ability to ask is the exception in every category rather than the rule. The read-only versus mutable split is the sharpest divide: CTabQA and CText2SQL never change the table, while manipulation and agentic systems must track a state their own actions alter, which is a harder context problem. Numerical chaining is concentrated in CTabQA finance datasets and is thin elsewhere. Initiative flips only in the agentic category, where the system, not the user, decides the next step. The connections below, and the cross-cutting analysis in Section 8, follow these axes.

**Foundations underpins the other categories.** Systems in categories 2 through 5 rely, explicitly or implicitly, on the table representation and serialization techniques developed in category 1. The choice of how to encode a table, whether through a specialised pretraining objective, a structured serialization format, or a retrieval-based compression scheme, affects performance across all downstream tasks. Improvements in Foundations therefore have broad leverage.

**CTabQA and CText2SQL serve the same user need differently.** Both categories address users who want to retrieve information from a table using natural language. The difference is in what the system produces: a natural language answer (CTabQA) or an executable SQL query (CText2SQL). In practice, these approaches are increasingly hybridized. Some recent systems for table QA and numerical reasoning generate intermediate SQL or Python code as a reasoning step before producing a natural language response (Cheng et al., 2023; Chen et al., 2022), blurring the categorical boundary.

**Interactive Manipulation adds a write dimension.** Categories 2 and 3 are read-only: the user queries a table that does not change. Category 4 introduces writing, where the table itself is modified. This changes the nature of dialogue context significantly, because the system must track not just what was asked and

answered but what the table currently looks like, a state that depends on all prior manipulation actions in the session.

**Agentic Systems subsume and extend.** An agentic system handling an open-ended data analysis goal may internally invoke querying, SQL generation, and table manipulation as sub-tasks within a single session. This makes agentic systems among the most powerful and the hardest to evaluate because, to our knowledge, no existing benchmark evaluates all of these capabilities within a single unified conversational framework.

**All categories share three cross-cutting challenges.** Regardless of whether the task is answering a question, generating SQL, editing a spreadsheet, or orchestrating a multistep analysis, every category must address: how to resolve ambiguous user intent, how to maintain and update dialogue context across turns, and how to measure progress when task definitions, data formats, and interaction modes vary across sub-areas. These three challenges are the subject of Section 8.

# 4 Conversational Table Question Answering

Conversational Table Question Answering (CTabQA) is the most direct expression of the survey's central problem: answering table-grounded questions whose interpretation depends on dialogue history. It covers systems that answer natural language questions grounded in tables or in hybrid documents that combine tables with prose text, where the questions form a dialogue in which the meaning of each question may depend on what was asked and answered before. This dependence is what separates CTabQA from single-turn table QA: a user does not issue one question and receive one final answer, but engages in an exchange that progressively builds an understanding of the data. The research challenges in CTabQA are qualitatively different from those in single-turn table QA. A system must track which entities, values, and columns have been mentioned across turns. It must interpret questions that omit information recoverable only from prior context. It must carry numerical results forward across turns rather than computing each answer independently. And in the most ambitious systems, it must decide when the user's intent is ambiguous enough to warrant asking a clarification question rather than guessing.

## 4.1 Background: From Text-Only Conversational QA to Tables

CTabQA inherited its formulation and evaluation from conversational QA over free text. CoQA (Reddy et al., 2019) and QuAC (Choi et al., 2018) established that multiturn QA is harder than single-turn QA and that single-turn models do not transfer to it. Moving to tables added three complications: tables are not linear, so position carries meaning; many questions need arithmetic; and hybrid documents require reasoning over table and prose together. On the table side, single-turn datasets set the stage: WikiTableQuestions (Pasupat & Liang, 2015) and FeTaQA (Nan et al., 2022) for table QA, and HybridQA (Chen et al., 2020b) and OTT-QA (Chen et al., 2021a) for multihop table-plus-text reasoning. CTabQA builds on all of this and adds the conversational dimension.

## 4.2 Datasets

Progress in CTabQA has been driven primarily by dataset construction. Each new dataset introduced one or more phenomena, such as clarification, numerical chaining, or hybrid grounding, that prior datasets did not cover. Table 11 provides a structured comparison of all CTabQA datasets discussed in this section.

The datasets share a small set of evaluation families: answer-level metrics such as exact match and token-level F1, program metrics such as execution accuracy and program accuracy for numerical reasoning, ranking metrics for retrieval, and text-generation metrics such as ROUGE, BLEU, and BERTScore. Execution accuracy checks whether a generated program runs to the correct answer; program accuracy checks whether the program itself matches the reference; BERTScore (Zhang et al., 2020) compares an answer to a reference using contextual embeddings rather than exact strings. Table 11 lists which families each dataset uses.

Rather than describe each dataset in turn, we read them as a progression and ask what the progression reveals.

**What each step added.** The datasets form a clear sequence, where each one introduced a phenomenon the earlier ones did not test. SQA (Iyyer et al., 2017) started the task by splitting one complex Wikipedia-table question into an ordered chain of simpler ones. HybriDialogue (Nakamura et al., 2022) added hybrid grounding, so an answer can come from the table or from the prose beside it. PACIFIC (Deng et al., 2022) added proactive clarification, making the decision to ask the user a modelled action rather than an afterthought. ConvFinQA (Chen et al., 2022) added numerical chaining, where a value computed in one turn becomes an unstated operand in a later one. MMCoQA (Li et al., 2022) added multimodal evidence across text, tables, and images. iTBLS (Sundar et al., 2025) and cPAPERS (Sundar et al., 2024) moved the setting to scientific tables, and iTBLS split the task into interpretation, modification, and generation, which is where table QA starts to overlap with manipulation. ConvFinQA and PACIFIC were built on the single-turn FinQA (Chen et al., 2021b) and TAT-QA (Zhu et al., 2021) datasets, which is why those two appear in Table 11 as predecessors.

**Why finance and Wikipedia dominate.** Two domains carry most of the datasets, for practical reasons. Wikipedia tables are public, plentiful, and easy to crowdsource over, so they suit early dataset building and general-domain QA. Financial reports are attractive for the opposite reason: they pair dense numerical tables with explanatory prose, the questions analysts ask need real multi-step arithmetic, and a wrong answer has a clear cost. Both domains offer tables that are either cheap to annotate or clearly worth getting right. Domains that lack one of these properties, such as medicine and law, are barely represented.

**What no dataset covers.** The gaps are as informative as the coverage. Most datasets ground each dialogue in a single table or single document, so multi-table, multi-document, and cross-period questions stay untested. The annotations score the final answer, so cell grounding, evidence selection, and the faithfulness of each numerical step are mostly unmeasured. Clarification appears in PACIFIC but is rare elsewhere, so it is not yet a standard expectation. Apart from cross-lingual work in CText2SQL, the datasets are English. We are not aware of a CTabQA dataset that brings several of these missing properties together.

**Which conventions are inherited habits.** Some evaluation choices reflect history more than fit. Exact match and sequence accuracy carry over from single-turn QA; exact match penalises correct paraphrases, and sequence accuracy marks a whole chain wrong for a single slip. Execution and program accuracy come from the FinQA line and assume the answer is the output of a generated program. Many "dialogues" are not naturalistic sessions but decompositions of one complex question, a convention that began with SQA and persists because it is cheap to annotate. These habits make scores easy to compare inside the category, but they also lock in a narrow view of what conversational table QA is.

### 4.3 Core Technical Challenges

Four challenges recur across CTabQA datasets, and each is the table-grounded form of a cross-cutting theme analysed in Section 8. *Context-dependent interpretation* requires tracking the entities, values, and time references from prior turns; concatenating the history as a text prefix works on short dialogues but degrades on long ones (Chen et al., 2022). *Coreference* points to cells by row and column rather than by name, so the prose entity-tracking that conversational QA relies on (Clark & Manning, 2016) does not transfer (Section 8.2.2). *Numerical chaining*, central to ConvFinQA and PACIFIC, reuses intermediate results across turns; current systems store them as named variables in generated code (Chen et al., 2022). *Proactive clarification* requires the system to judge its own uncertainty, which among the datasets we reviewed only PACIFIC makes an explicit objective (Deng et al., 2022).

### 4.4 Modelling Approaches

CTabQA models evolved from task-specific pipelines toward LLM-based systems, tracking both the maturation of language models and the growing complexity of the datasets.

**Retrieval-then-generate.** HybriDialogue introduced the retrieval-then-generate pattern: retrieve the relevant table (and a passage, in hybrid settings), then generate the answer from the retrieved evidence and the

history (Nakamura et al., 2022). cTBLS (Sundar & Heck, 2023) instantiates it with dense table retrieval, coarse-to-fine cell ranking, and an LLM generator, and reports large retrieval and response-quality gains over a BM25 baseline on HybriDialogue.

**Task-specific baselines.** Two dataset baselines mark the pre-LLM approach. UniPCQA (PACIFIC) is a sequence-to-sequence model trained jointly on answering and clarification, and recasts numerical reasoning as code generation (Deng et al., 2022). TAGOP (TAT-QA) tags relevant tokens in the table-text input and applies an operator, which works for single-turn numerical QA but does not extend to multiturn settings where the relevant values change (Zhu et al., 2021).

**LLM-based systems.** In-context learning shifted the field from task-specific architectures to prompted general-purpose models. EHRAgent (Shi et al., 2024) shows the pattern in medicine: it answers questions over electronic health records by generating Python that queries the database, which separates understanding from computation and avoids arithmetic hallucination. More broadly, recent systems on SQA and ConvFinQA rely on prompting, retrieval, or program execution rather than bespoke architectures (Sui et al., 2024; Khatuya et al., 2025), which is why the strongest current systems are usually LLM-mediated even though the category's core datasets predate the LLM wave.

## 4.5 Domain-Specific Directions

CTabQA research clusters in two domains, for practical reasons. Finance has the densest set of datasets (TAT-QA, FinQA, ConvFinQA, PACIFIC, and MultiHiertt (Zhao et al., 2022), the last over hierarchical tables with subtotals), because financial reports pair numerical tables with prose, the questions need multistep arithmetic, and errors are costly. Science is the other focus: iTBLS (Sundar et al., 2025) and cPAPERS (Sundar et al., 2024) target experimental result tables from arXiv papers, whose abbreviated columns need domain knowledge to read. Medicine appears through EHRAgent (Shi et al., 2024) over electronic health records, where privacy and the cost of errors dominate; most medical QA benchmarks such as MedQA (Jin et al., 2020) remain single-turn and are not grounded in EHR tables.

## 4.6 Category-Specific Remaining Gaps

The unresolved issues in CTabQA are now fairly specific to the task formulation itself. Most benchmarks still assume a single table or single document per dialogue, so the category remains weak on multi-table, multi-document, and cross-period analysis. The strongest datasets also concentrate on answer correctness, leaving cell grounding, supporting evidence selection, and stepwise numerical faithfulness under-measured. Clarification appears explicitly only in a small part of the literature, most notably PACIFIC, rather than as a standard expectation for table-grounded QA.

# 5 Conversational Text-to-SQL

Conversational Text-to-SQL (CText2SQL) has the most standardized evaluation setup of the five categories. We judge this by three concrete signals: two shared benchmarks that most systems report on (SParC and CoSQL), hidden test splits, and a public leaderboard. The task is to translate a user's natural language question into a SQL query (a formal expression in Structured Query Language used to retrieve data from a relational database), where the correct translation of the current question may depend on the SQL queries generated in prior turns of the same dialogue.

The category also has a clear benchmark progression and an architecture trajectory that moves from hand-crafted SQL-editing rules through graph-based neural encoders to large language model prompting. This shared setup makes CText2SQL the easiest category to compare across systems. It also means its benchmarks and evaluation conventions have settled in ways that limit generalization to real-world settings.

## 5.1 Background: From Single-Turn to Multiturn Text-to-SQL

Text-to-SQL predates deep learning: the ATIS benchmark (Dahl et al., 1994) established the single-domain setting over air-travel queries, but models trained on it did not generalize to new databases. Spider (Yu et al., 2018) shifted the field to cross-domain Text-to-SQL, with a split that guarantees no database appears in both training and test, and single-turn systems improved quickly on it, from relation-aware encoders (Wang et al., 2020; Scholak et al., 2021; Cao et al., 2021) to few-shot LLM prompting (Gao et al., 2024; Pourreza & Rafiei, 2023), reaching about 86.6% execution accuracy against an estimated 92.6% human ceiling. CText2SQL extends this to the realistic case where users ask a sequence of related questions, and its datasets SParC and CoSQL are built on the Spider databases so single-turn comparison stays possible. Surveys of single-turn Text-to-SQL include Katsogiannis-Meimarakis & Koutrika (2023) and Liu et al. (2024); we focus on the multiturn extension.

## 5.2 Datasets

Two benchmark datasets define the core CText2SQL landscape: SParC and CoSQL. Both are built on the Spider database collection, which enables direct comparison with single-turn Text-to-SQL baselines and isolates the additional difficulty introduced by conversational context. For reference, Table 12 also includes historically important or practically relevant non-conversational Text-to-SQL benchmarks discussed in this section.

**SParC.** SParC (Semantic Parsing in Context, Yu et al., 2019b) is the primary CText2SQL benchmark. It contains 4,298 interaction sequences derived from Spider questions, with 12,166 individual questions spread across 200 databases. Expert annotators constructed the sequences by designing question chains that progressively explore a database topic: a sequence might begin with a broad question about all employees, then narrow to a department, then filter by salary, then ask for a count. Each question in a sequence was annotated with its gold SQL, making SParC suitable for turn-level evaluation.

SParC is evaluated using two metrics. Question match (QM) measures Spider-style exact set match at the turn level: a turn is counted as correct only if the predicted SQL matches the gold SQL under the benchmark's clause-level exact-match evaluator, not by raw string identity. Interaction match (IM) is stricter: a full interaction is counted as correct only if every turn in it achieves QM. IM directly measures end-to-end conversational correctness but is so strict that most systems score well below 50% on it even when turn-level accuracy is substantially higher.

**CoSQL.** CoSQL (Yu et al., 2019a) was collected using the Wizard-of-Oz protocol applied to the Spider databases. Pairs of crowd workers were recruited: one played the role of a user exploring a database, and the other acted as a SQL expert with full access to the database schema, answering questions by writing and executing SQL queries. The user could not see the database schema (the formal description of table names and column names) and had to ask questions in natural language; the system worker provided answers and, when necessary, asked clarification questions. This asymmetric setup produces dialogues that are more naturalistic than SParC's expert-designed sequences because the user's questions emerge from genuine information-seeking rather than from a pre-planned decomposition.

CoSQL adds two tasks beyond SQL state tracking: response generation from database results and user dialogue act prediction. This makes CoSQL historically important within CText2SQL because it evaluates not only whether the system recovers the correct SQL, but also whether it can communicate appropriately during database interaction.

**BIRD and Spider 2.0.** BIRD (Li et al., 2023b) is a single-turn benchmark but is widely used in the CText2SQL literature as a measure of difficulty relative to Spider. Its 12,751 questions are more complex than Spider's, requiring external knowledge (facts not present in the database schema) and involving larger, dirtier databases. BIRD's difficulty ceiling serves as a reference for evaluating how close CText2SQL systems are to handling realistic enterprise queries.

Spider 2.0 (Lei et al., 2025) takes this realism further by using actual enterprise database environments involving multi-database workflows, nested queries, and operations that go beyond standard SELECT statements. Neither BIRD nor Spider 2.0 is multiturn, but both signal that the gap between current CText2SQL benchmarks and real-world database interaction remains large.

### 5.3 Core Technical Challenges

CText2SQL inherits the single-turn Text-to-SQL challenges and adds ones specific to dialogue. *Schema linking shifts across turns*: the relevant tables and columns change as the user moves focus, and a turn like "now look at the managers instead" changes the target table without naming it, so the system must track what is in and out of scope. *SQL context*: successive queries usually differ by a clause or two, so a system that regenerates SQL from scratch each turn produces the wrong incremental edit. This is what the SQL-editing models in Section 5.4 target. *Implicit reference and ellipsis*: a turn like "what about in 2022?" omits the attribute, the aggregation, and the table, all recoverable only from state. *Clarification and error propagation* are the table-grounded forms of the cross-cutting themes in Section 8: CoSQL shows that users ask schema-ambiguous questions, and a wrong early SQL misleads every dependent later turn, which motivates explicit interaction-state tracking (Wang et al., 2021).

### 5.4 Modelling Approaches

CText2SQL models have evolved through three identifiable generations. Table 13 summarises the key systems discussed in this section.

**Generation 1: SQL-editing models (2019 to 2021).**  The earliest systems treated the current question as an edit of the prior SQL query. Edit-SQL (Zhang et al., 2019) made this explicit: instead of generating each turn from scratch, it conditions on the previous SQL and predicts the edits that turn it into the current query. This matches the intuition that adjacent turns usually differ by a few clauses. The weakness is that it depends on the prior SQL being right, so an early error propagates forward.

**Generation 2: Graph-based encoders (2020 to 2022).**  The second generation moved the effort from the output side to the input side, encoding the schema and the dialogue history together. IGSQL (Cai & Wan, 2020) added a schema interaction graph that links schema items referenced together across turns. HIE-SQL (Zheng et al., 2022) added a history-enhanced encoder that models coreference between the current question and prior turns. IST-SQL (Wang et al., 2021) kept an explicit interaction state of which schema elements, filters, and aggregations are in scope. RASAT (Qi et al., 2022) is the strongest of this generation: it builds on the T5 sequence-to-sequence model and injects cross-turn coreference as explicit relations in the attention mechanism, and it set the state of the art on SParC and CoSQL at the time.

**Generation 3: Large language model approaches (2022 to present).**  The third generation prompts a general-purpose LLM instead of fine-tuning a task-specific model. ACT-SQL (Zhang et al., 2023a) showed that a prompt holding the schema, the dialogue history, and a few examples is enough to generate competitive SQL with no task-specific training. CoE-SQL (Zhang et al., 2024a) ports the Generation 1 idea into the prompt: it asks the model to write each turn's SQL as an edit of the previous one. Track-SQL (Chen et al., 2025a) splits the work across two extractors, one for schema and one for context-dependent tokens, before generation.

**Reading the exact-match numbers in Table 13.**  On exact set match (QM and IM), the fine-tuned Generation 2 models score higher than the LLM-prompting systems: RASAT reaches 67.7 QM on SParC, against 56.0 for CoE-SQL and 51.0 for ACT-SQL. This ordering should be read with care, for three reasons. First, QM and IM compare the predicted SQL to the gold SQL clause by clause, so they reward writing the query in the same form the annotators used. Fine-tuned models are trained on that gold SQL, while a prompted LLM often writes SQL that returns the right answer in a different form, which exact match marks wrong. Second, the rows do not all use the same split: the RASAT and HIE-SQL numbers are official test results, while the ACT-SQL, CoE-SQL, and Track-SQL numbers are development results. Third, when

CoE-SQL is scored by execution rather than exact match, the same system reports 74.1 question and 51.9 interaction accuracy on SParC, much closer to the fine-tuned models. So the exact-match gap reflects what the metric and the split measure as much as a real difference in SQL quality, a point we return to in Section 8.3.

**Multi-agent and search-based approaches.** A separate line decomposes generation across agents or search. MAC-SQL (Wang et al., 2025) uses a selector, a decomposer, and a refiner; its decomposition mirrors the structure of a multiturn session, even though it was built for single-turn SQL. CHASE-SQL (Pourreza et al., 2025) uses multi-path reasoning with preference optimization, and Alpha-SQL (Li et al., 2025) treats SQL construction as a search problem using Monte Carlo Tree Search. Both are evaluated mainly in single-turn settings, but their structure extends naturally to conversational SQL.

Viewed across these modelling families, CText2SQL also has a useful, but limited, connection to dialogue state tracking (DST). The analogy helps explain the design space without defining a separate modelling lineage: SQL-editing systems treat the next turn as a state update over the previous query; question-rewriting systems externalise context resolution by converting an elliptical turn into a self-contained one; and IST-SQL (Wang et al., 2021) comes closest to explicit state maintenance over a structured interaction history. What is still missing is a widely adopted query-state representation that separates confirmed constraints from inferred ones and supports repair when later turns reveal that an earlier assumption was wrong. We therefore use the DST analogy here as an interpretive lens on the models above, and return to its broader cross-cutting significance in Section 8.2.3.

## 5.5 Data Augmentation and Cross-Lingual Extensions

A persistent limitation of CText2SQL research is data scarcity. Constructing multiturn Text-to-SQL datasets requires annotators who understand both natural language and SQL, and who can design interaction sequences that test a specific conversational phenomenon. This expertise is expensive, which is why SParC and CoSQL together contain fewer than 8,000 dialogues, a much smaller training set than comparable single-turn benchmarks.

Liu et al. (2022a) introduced self-play data augmentation as a response to this scarcity. A SQL-to-text model (a model that generates natural language questions from SQL queries) and a Text-to-SQL model are alternated: the SQL-to-text model generates new questions for unseen databases, and the Text-to-SQL model generates their SQL annotations. This bootstrapping procedure produces training data for new databases without requiring human annotators, and the models trained on the augmented data generalize better to new database schemas.

Question rewriting (Vakulenko et al., 2021) offers a complementary approach. A context-dependent question such as "and what about the previous year?" is rewritten into a fully self-contained question that incorporates the missing context explicitly, such as "what was the total revenue in 2021?" The rewritten question can then be handled by any single-turn Text-to-SQL system without modification. Question rewriting separates the conversational context resolution problem from the SQL generation problem, making it easier to plug in improved single-turn SQL generators as they appear. Its limitation is that the rewriter itself must handle coreference and ellipsis, shifting the difficulty rather than eliminating it.

Cross-lingual extensions illustrate another open front for CText2SQL. Most existing work focuses on English, but database users worldwide issue queries in many languages. The structural challenges of multiturn SQL generation therefore compound with the linguistic challenges of cross-lingual transfer (applying a model trained primarily on one language to queries in another language).

## 5.6 Category-Specific Remaining Gaps

CText2SQL is comparatively mature, but its main remaining gaps are tightly tied to semantic parsing over realistic databases. Its benchmarks still underrepresent messy enterprise schemas, incomplete metadata, and deployment settings where schema exposure, privacy, and access control matter. A second local weakness is that clarification is not yet deeply integrated into SQL state tracking: systems can often recover the next

query, but they are less reliable at deciding when ambiguity warrants asking the user and then constraining later SQL with the answer. Finally, strong per-turn generation still does not fully solve interaction-level consistency across long query sequences.

## 6  Interactive Table Manipulation

The previous two sections covered systems that *read* tables: they answer questions or generate SQL in response to user queries, but the underlying table does not change. Interactive Table Manipulation covers a qualitatively different mode of interaction: the user instructs the system to *change* the table. The output of a successful interaction is not an answer but a modified table, a new visualization, or a generated piece of table content that reflects the user's intent.

This category encompasses two lines of research that developed largely independently. The first, rooted in NLP and human-computer interaction (HCI), addresses spreadsheet agents: systems that accept natural language instructions and translate them into sequences of operations that modify a spreadsheet. The second, rooted in the visualization community, addresses natural language interfaces for data exploration: systems that allow users to iteratively refine data visualizations through natural language. Both lines share the property that interaction is genuinely bidirectional: the user's input changes the artifact (spreadsheet or visualization) that the system displays, and subsequent user inputs are interpreted in the context of that changed artifact.

### 6.1  Background: Natural Language Interfaces and HCI

Controlling software through natural language is decades old: early database interfaces such as LUNAR (Woods, 1973) and INTELLECT used hand-crafted, brittle grammars. The problem is harder than Text-to-SQL because a general application's action space is far larger and less formal than the SQL grammar. Large language models reopened it, since models trained on code can generate function calls for Excel, Google Sheets, or pandas without hand-written grammars, which drove the expansion of spreadsheet agents since 2023. On the visualization side, the HCI community built interactive natural language interfaces such as Eviza (Setlur et al., 2016) and NL4DV (Narechania et al., 2021) before the LLM wave; their user-study evaluation differs from the benchmark-driven NLP style, which makes cross-community comparison hard.

### 6.2  The Manipulation Task Space

Interactive table manipulation can be organised along two dimensions: the *target* of the operation and the *type* of operation applied.

**Targets** include cells, rows, columns, worksheets, charts, pivot tables, and, in transformation settings, intermediate or output tables. The same natural language instruction can imply different executable actions depending on the target. For example, "remove duplicates" applied to a column means deduplicating values within that field, whereas the same instruction applied to rows means deleting repeated records.

**Operation types** include entry (setting or clearing values), management (inserting, deleting, reordering, or creating structural units such as rows, columns, and sheets), formatting (changing appearance or display format), formula writing (inserting spreadsheet formulas), data transformation (reshaping or synthesising tables, including pivoting, unpivoting, melting, or merging), and chart generation (creating or modifying visualisations from tabular data).

Table 14 summarises representative systems in this space. The landscape is still fragmented. General-purpose spreadsheet agents cover parts of entry, management, formatting, charting, or formula-related work, but broad end-to-end support remains limited, especially for long-horizon tasks and transformation-heavy workflows. By contrast, systems such as Rigel and NL2Rigel specialise in table restructuring, while Eviza, NL4DV, and Orko focus primarily on chart specification or interactive visual analysis rather than cell-level spreadsheet editing.

### 6.3 Benchmarks

Unlike CTabQA and CText2SQL, which converged on a small number of community-standard benchmarks, Interactive Table Manipulation has a fragmented evaluation landscape. Each major system introduced its own benchmark, making direct comparison across systems difficult.

**SheetCopilot Benchmark (SCB).** The SheetCopilot Benchmark (SCB) contains 221 spreadsheet control tasks distributed across 28 Excel workbooks, derived from spreadsheet-related Q&A (e.g., SuperUser) and adapted to a fixed workbench (Li et al., 2023a). To analyse task diversity, SheetCopilot defines six categories: entry and manipulation, management, formatting, charts, pivot tables, and formulas. Each task specifies a starting spreadsheet state and a natural language instruction; the evaluation executes the generated action sequence and compares the resulting sheet state against one of the ground-truth solutions using an automated comparator.

SCB reports multiple metrics. Exec@1 measures whether the generated action sequence executes without throwing exceptions, while Pass@1 measures functional correctness of the final spreadsheet state. Sheet-Copilot also reports efficiency metrics (A50 and A90), which summarise the ratio between the number of generated atomic actions and the number in a reference solution.

**SheetRM.** SheetRM (Chen et al., 2025b) was introduced alongside SheetAgent to better reflect real-life spreadsheet manipulation challenges. Compared with SCB, which primarily targets short-to-medium horizon tasks, SheetRM emphasises long-horizon, multi-category tasks where later steps depend on earlier intermediate results and where choosing the correct operation often requires reasoning over spreadsheet content. In addition to Exec@1 and Pass@1, SheetRM reports SubPass@1, which measures the fraction of subtasks completed correctly within a multi-step task and helps distinguish partial progress from total failure.

**SpreadsheetBench.** SpreadsheetBench (Ma et al., 2024) is a challenging benchmark introduced at NeurIPS 2024 (Datasets and Benchmarks Track). It contains 912 real instructions gathered from four online Excel forums (ExcelForum, Chandoo, MrExcel, and ExcelGuru), paired with spreadsheets that reflect real-world complexity (e.g., multiple tables, non-standard relational layouts, and non-textual elements). To reduce false positives from solutions overfitting a single spreadsheet instance, SpreadsheetBench adopts an online-judge style protocol: each instruction is evaluated on multiple input–output test cases (2,729 total, ~3 per instruction), and a solution is counted as correct only if it passes all associated cases.

The benchmark exposes a large gap between current agents and human performance. In the authors' evaluation, even strong systems such as Copilot in Excel achieve only around 20% accuracy, and GPT-4o scores around 17% under single-round inference, indicating that realistic formula writing and transformation-heavy tasks remain far from solved.

**Visualization benchmarks.** The visualization systems in this category are evaluated through user studies and analytic specification accuracy rather than programmatic task completion. NL4DV (Narechania et al., 2021) evaluates accuracy of the analytic specification (a formal description of what visualization to produce, including the data attributes to map to visual channels such as x-axis, y-axis, and color) generated from a natural language query. Eviza (Setlur et al., 2016) and Orko (Srinivasan & Stasko, 2018) were evaluated primarily through user studies measuring task completion time, error rate, and subjective satisfaction. The absence of a shared programmatic evaluation standard for natural language driven visualization is a recurring limitation noted in the visualization literature.

### 6.4 Spreadsheet Manipulation Systems

These systems share a common design: a library of spreadsheet actions, plus a loop in which the model proposes the next action, sees the result, and revises. Table 14 lists their operations, backbones, and benchmarks. They differ along a few axes that matter more than their individual feature lists: how they

plan, whether they use one agent or several, whether the backbone is proprietary or open, and whether the user specifies the steps or the goal.

**From a single loop to modular planning.** SheetCopilot (Li et al., 2023a) set the template: an atomic action library used as a virtual API, driven by a state-machine loop that proposes, revises, and acts on a serialized view of the sheet. It reaches 44.3% Pass@1 on its own benchmark (SCB) with GPT-3.5. SheetAgent (Chen et al., 2025b) keeps the loop but splits the work across a Planner, an Informer that retrieves the relevant sheet context, and a Retriever that maps the plan to API calls. The split lowers the context load per step and raises Pass@1 on SCB to 61.1%. The same paper introduces SheetRM, a long-horizon benchmark, where SheetAgent reaches only 44.8% Pass@1 even though 77.0% of its subtasks are solved. That gap between subtask and task success is the clearest sign that long, dependent workflows are the open problem in this category.

**Variations on the agent design.** Later systems vary the same template. SheetMind (Zhu et al., 2025) uses three agents and constrains the action agent with a Backus–Naur Form grammar, so generated actions are well-formed by construction, with a reflection agent that checks intent before execution. SpreadsheetLLM (Tian et al., 2024) is not an agent at all: it is a compression scheme that fits large sheets into a context window by keeping anchor cells and encoding type patterns instead of raw values, which any of these agents can use as a front end. TableLLM (Zhang et al., 2025) swaps the proprietary backbone for a fine-tuned open one (Llama 3.1-8B), so sensitive office documents need not leave the user's machine, trading some generality for privacy.

**Declarative transformation.** Rigel (Chen et al., 2023) and NL2Rigel (Huang et al., 2024a) take a different stance on reshaping tasks such as pivoting and joining. Instead of asking the model to emit imperative steps, the user describes the shape of the wanted output and the system infers the reshape operations. NL2Rigel adds an LLM front end and an interface that shows the pipeline and intermediate results, so the user can correct it across turns. In a user study it matched Rigel's completion rate (86 of 96 tasks) at lower time cost, which suggests the declarative framing helps most when the transformation is hard to spell out step by step.

## 6.5 Interactive Visualization and Data Exploration

The visualization sub-community developed natural language interfaces for data exploration independently of the NLP spreadsheet agent literature, with a different set of technical approaches and evaluation conventions. These systems share the core conversational property that each user utterance is interpreted relative to the current visualization state, but they focus on the chart rather than the underlying table as the artifact being modified.

**Eviza.** Eviza (Setlur et al., 2016) is one of the earliest conversational natural language interfaces for visual data analysis, published at UIST 2016. It allows users to issue natural language queries about a scatterplot or bar chart and responds by updating the visualization: filtering points, changing the axis variable, or highlighting a subset of data. A key design contribution is Eviza's handling of pragmatic ambiguity (ambiguity that arises not from the words themselves but from context and conversational implicature, which is the unstated meaning that a speaker intends a listener to infer). For example, the query "show me the expensive ones" is pragmatically ambiguous because "expensive" is relative; Eviza resolves this by asking the user to confirm its interpretation before applying the filter. This proactive disambiguation prefigures the clarification mechanisms studied in PACIFIC and CoSQL.

**NL4DV.** NL4DV (Natural Language for Data Visualization, Narechania et al., 2021) is a developer-facing toolkit that translates natural language queries about a tabular dataset into analytic specifications expressed in the Vega-Lite grammar (a JSON-based language for describing data visualizations). NL4DV operates through a pipeline: it identifies data attributes mentioned in the query, infers the intended analytic task (comparison, distribution, correlation, and so on), selects an appropriate chart type, and generates the corresponding Vega-Lite specification. The original NL4DV paper primarily targets one-shot utterances;

conversational behaviour is typically implemented by the surrounding application by maintaining and editing prior specifications across turns.

**Orko.** Orko (Srinivasan & Stasko, 2018) extends the conversational natural language interface paradigm to network visualizations: graphs of nodes and edges such as social networks, citation networks, or molecule structures. Users can issue natural language commands to filter nodes, highlight paths, change the layout, or query properties of the network. Orko is relevant to this survey because its underlying technical challenges (tracking conversational context over a structured visual artifact, resolving references to previously selected nodes) are structurally analogous to those in table manipulation, even though the target artifact is a network rather than a table.

## 6.6 Core Technical Challenges

Manipulation adds challenges that querying and translating do not have. *Mutable state*: the table changes with every operation, so the system must track both what was said and what was done, a harder problem than tracking dialogue over a fixed table (Section 8.2.2). *Action vocabulary*: an instruction must map to operations from an application-specific library, and the choice between generating raw code (general but unsafe, since it can corrupt data) and a fixed action library (safe but limited) is a real design tension. *Underspecification*: an instruction like "clean up this table" admits a huge space of actions, and most systems just pick a default, which fails on complex tasks. *Verification and safety*: destructive edits can cause irreversible data loss, so the system should check the result before continuing; benchmarks report executability and correctness, but explicit confirmation loops for high-risk edits are rare. *Long-horizon planning*: a request like "create a monthly revenue summary by product category" expands into a dozen operations, which is what SheetRM stresses; on it SheetAgent reaches 44.8% Pass@1 with 77.0% SubPass@1 (Chen et al., 2025b), so subtasks are often solved even when full completion is not.

## 6.7 Category-Specific Remaining Gaps

The bottlenecks in interactive manipulation are distinctive because the system acts on mutable artifacts rather than only answering questions. Current benchmarks still underrepresent the structural messiness of real spreadsheets, including merged cells, multilevel headers, hidden formulas, embedded charts, and irregular layouts. The category is also unusually dependent on safeguards: confirmation, rollback, provenance, and other protections against destructive edits are central requirements here rather than optional interface features. Evaluation therefore needs to capture not just task completion, but whether the right edits were made safely to the evolving workbook state.

# 7 Agentic Table Systems

The four preceding categories all assume that the user controls the granularity of each step. In CTabQA the user asks a question; in CText2SQL the user issues a query; in Interactive Manipulation the user gives an instruction. In every case the user decides what to do next. Agentic Table Systems invert this arrangement. The user states a high-level goal and the system decides autonomously how to decompose that goal, which tools to invoke, in what order, and how to handle the results of intermediate steps.

This shift from user-directed to system-directed task execution is what the term *agentic* refers to in the context of LLM-based systems. An agent in this sense is a language model that perceives inputs from its environment (the current table state, prior tool outputs, the user's goal), reasons about what action to take next, executes that action through a tool call, observes the result, and continues this loop until the goal is achieved or the system determines it cannot be achieved. This perception-action-observation loop, often called the ReAct pattern (Reasoning and Acting, Yao et al., 2023), is the architectural foundation of many agentic table systems.

Agentic systems are among the most powerful and hardest-to-evaluate categories in this survey. They are among the most powerful because they can subsume all four preceding interaction modes within a single session: a user goal such as "prepare a summary of our quarterly revenue performance" may require the agent

to query tables (CTabQA), generate SQL (CText2SQL), edit a spreadsheet (Interactive Manipulation), and compile a report, all without the user issuing separate instructions for each step. They are hard to evaluate because, to our knowledge, no single benchmark currently tests all of these capabilities within a unified conversational framework, and because the open-ended nature of the task makes it difficult to define what a correct response looks like.

## 7.1 Background: Agentic LLM Infrastructure

Agentic table systems build on general-purpose agent techniques. Chain-of-thought prompting (Wei et al., 2022) elicits intermediate reasoning, used here to decompose a goal and choose tools. Retrieval-augmented generation (Lewis et al., 2020) fetches relevant data so the agent need not hold every table in context; TableRAG (Chen et al., 2024) applies this to large tables. Tool-augmented reasoning (Schick et al., 2023) lets the model call calculators, code interpreters, and database connectors, which matters because arithmetic and SQL are better handled by deterministic tools than by the model. Multi-agent frameworks (Hong et al., 2024) split a task across specialised agents that communicate, often along functional lines: one plans, one queries, one verifies, one synthesises.

## 7.2 Systems and Benchmarks

Table 15 summarises representative agentic systems, prototypes, and benchmarks discussed in this section.

The systems in Table 15 differ less in the tasks they perform than in how they organize the work. Read together, they trace a few recurring design choices: how planning is structured, whether one agent or several do the work, whether the database is a source or a memory, and whether the user stays in the loop.

**Planning and workflow.** Data-Copilot (Zhang et al., 2023b) is one of the earliest end-to-end systems and makes planning explicit: it first synthesises a full workflow from the user's goal, then executes it, which separates reasoning about the task from carrying it out. Its released version runs over heterogeneous financial sources such as stocks, funds, and macroeconomic indicators. DS-STAR (Nam et al., 2025) pushes the same plan-then-execute idea further with specialised agents for planning, routing, execution, and verification, and reports gains over earlier DA-Code baselines. The trend is away from one prompt that emits code and toward an explicit, inspectable plan.

**Multi-agent reasoning and critique.** Several systems split the work across agents. AutoTQA (Zhu et al., 2024) runs tabular QA through a planner-executor team that works across multiple tables and databases, so the system, not the user, decomposes the question. Table-Critic (Yu et al., 2025) adds a critic agent that reviews answers and asks for revisions, a self-refinement loop that improves results on WikiTableQuestions and TabFact. TALON (Jin et al., 2025) combines planning, tool use, and critique for question answering over long tables. The shared bet is that a second agent checking the first is cheaper than getting the first right in one pass.

**The database as memory, not just source.** Two systems re-cast the database itself. ChatDB (Hu et al., 2023) uses the database as symbolic memory: the agent writes intermediate facts to tables and reads them back with SQL, which lets it hold structured state across a long session, much like dialogue state tracking at larger scale. D-Bot (Zhou et al., 2024) turns the agent on the database system, diagnosing performance faults by reading execution plans and logs, which stretches "interaction with structured data" to database administration over system metadata.

**Benchmarks for agentic data work.** DA-Code (Huang et al., 2024b) and InfiAgent-DABench (Hu et al., 2024) evaluate agents rather than propose them. Both grade whether generated code executes to the right result instead of whether it matches a reference string, which is the right target for code-writing agents. DA-Code mixes data wrangling, analysis, and model training; InfiAgent-DABench focuses on the analysis step.

**Keeping the user in the loop.** Most of these systems run autonomously, which is the defining trait of the category but also its main risk on ambiguous tasks. Harmonia (Santos et al., 2025) is the clearest counterexample: it treats data harmonization, reconciling records that use different conventions, as a conversation, asking the user when a decision is ambiguous instead of guessing. EHRAgent (Shi et al., 2024) applies the code-generating agent to electronic health records, where a question spans several linked tables and errors are costly. Harmonia's conversational stance is the bridge back to the rest of this survey: it is where agentic autonomy meets the clarification problem of Section 8.1.

## 7.3 Key Design Dimensions

The systems described above differ along several dimensions that are worth making explicit, because these dimensions define the design space for future agentic table systems.

**Single-agent versus multiagent.** Some systems use a single LLM that handles all aspects of the task within one reasoning loop. Others decompose the task among multiple specialised agents. Multiagent designs can be more robust because each agent can be optimised for its sub-task, but they are more complex to coordinate and debug. The critic-reviser pattern used in Table-Critic and the planner-executor pattern used in AutoTQA represent two common multiagent decompositions. Data-Copilot, by contrast, is better viewed as a single-agent system with a plan-and-dispatch workflow.

**Plan-then-execute versus step-by-step.** Some systems synthesise a complete plan before executing any action (Data-Copilot). Others interleave reasoning and execution, deciding the next action only after observing the result of the previous one (the ReAct pattern). Plan-then-execute is more efficient when the goal is well-specified but fails when the plan cannot anticipate intermediate results. Step-by-step execution is more adaptive but can get stuck in loops or lose track of the overall goal in long sessions.

**Tool repertoire.** Systems vary in which tools they can invoke: SQL execution engines, Python code interpreters, spreadsheet APIs, web search, and file I/O. A richer tool repertoire enables more tasks but increases the difficulty of tool selection (deciding which tool to use at each step) and tool composition (combining multiple tools correctly in a single workflow).

**Conversational integration.** Some agentic systems are designed to operate entirely autonomously, without user interaction during execution. Others integrate conversational turns into the execution loop, pausing to ask for clarification when the goal is ambiguous or when an intermediate result is unexpected. Harmonia is the clearest example of conversational-agentic integration; most other systems in this category are predominantly autonomous. This dimension is the primary connection between the agentic category and the conversational themes that unite this survey.

**Data heterogeneity.** Some systems operate on a single structured data format (relational database, spreadsheet). Others handle heterogeneous data: combining structured tables with unstructured text, time-series data, images, or external API data. Data-Copilot and InfiAgent-DABench represent the heterogeneous end of this spectrum; D-Bot and AutoTQA represent the structured-only end.

## 7.4 Relationship to the Preceding Categories

Agentic table systems do not replace the preceding four categories; they build on them. The taxonomy survey by Tian et al. (2026) identifies five competencies that a complete table agent must possess: structure understanding (knowing how to encode and interpret tables), semantic understanding (knowing what a user's goal means), retrieval and compression (knowing how to find the relevant data within a large collection), executable reasoning (knowing how to produce and execute correct code or SQL), and cross-domain generalization (performing reliably on tables from domains not seen during training). Each of these competencies maps onto one or more of the preceding categories: structure understanding maps to Foundations, semantic understanding maps to CTabQA, executable reasoning maps to CText2SQL and Interactive Manipulation, and retrieval and compression maps to techniques used across all four categories.

What the agentic category adds is *orchestration*: the ability to invoke these competencies in sequence, coordinate their outputs, and maintain coherence across a session that may involve dozens of steps. In the

current literature, no system appears to achieve all five competencies reliably across diverse real-world data environments, and the evaluation frameworks needed to measure all five simultaneously do not yet exist.

## 7.5 Category-Specific Remaining Gaps

Agentic table systems face a distinctive evidence problem: they perform long tool-using workflows, but are rarely evaluated under sustained conversational supervision. Most benchmarks still test task completion more than interactive planning quality, intermediate recovery, or the visibility of provenance across generated code, SQL, search, and edits. The category also remains closer to autonomous batch execution than to true back-and-forth analytical partnership, so user intervention, correction, and mixed-initiative control are still underdeveloped. In this category especially, trustworthy orchestration is part of the core task, not an afterthought.

# 8 Cross-Cutting Themes

The five categories surveyed in the preceding sections address different interaction modes and different data formats, but they face a common set of challenges that do not belong to any one category. These challenges arise wherever a system must sustain a coherent interaction with a user over multiple turns of a conversation grounded in structured data. We identify three such cross-cutting themes: intent disambiguation and clarification (Section 8.1), dialogue context tracking (Section 8.2), and evaluation and benchmarks (Section 8.3). Together they explain why otherwise different categories often fail in the same ways. These themes are where the capability axes of Table 1 turn concrete: clarification is the "ask" axis, context tracking is where the read-only versus mutable state and numerical-chaining axes bite, and evaluation is what fails to measure any of them across a full dialogue. Understanding these themes together, rather than within individual category sections, serves two purposes. First, it reveals structural parallels that are invisible when each category is treated in isolation: the way PACIFIC handles clarification in CTabQA and the way CoSQL handles clarification in CText2SQL are conceptually the same mechanism applied in different settings, yet the two research communities have usually been discussed and evaluated separately. Second, it identifies the gaps that recur across otherwise fragmented literatures, allowing this section to synthesise them once before Section 9 turns that synthesis into a concise final agenda.

## 8.1 Intent Disambiguation and Clarification

Every conversational system grounded in structured data must at some point encounter a user input whose meaning is not uniquely determined by the available context. The input may be ambiguous because the user used a word that maps to multiple database columns, because the user omitted information that the system needs to generate a correct answer, or because the user's goal could be satisfied in more than one way and the user has not indicated a preference. The question of what a system should do in this situation is the clarification problem.

### 8.1.1 Forms of Ambiguity in Table Interactions

Ambiguity in table-grounded dialogue takes several distinct forms, each requiring a different resolution strategy.

**Lexical ambiguity** arises when a word or phrase in the user's input maps to more than one column, table, or value in the data. A user who asks "what are the revenues?" over a database that contains both gross revenue and net revenue columns has issued a lexically ambiguous question. The system must either choose one interpretation, ask the user to specify, or return both.

**Referential ambiguity** arises when a pronoun or demonstrative expression could plausibly refer to more than one entity introduced in the prior dialogue. A question such as "how did it change year over year?" is referentially ambiguous if the prior turns introduced multiple numerical quantities.

**Scope ambiguity** arises when the extent of a user's request is unclear. A question such as "show me all the recent transactions" is scope-ambiguous because "recent" is not defined: it could mean the last day, the last

month, or the last year. Scope ambiguity is particularly common in financial and scientific table interactions where temporal and numerical ranges are frequently left implicit.

**Underspecification** is a more radical form of ambiguity in which the user's instruction is compatible with a very large number of different actions. As discussed in Section 6.6, an instruction such as "clean up this table" is so underspecified that no single correct interpretation exists. Underspecification is most common in Interactive Table Manipulation.

### 8.1.2   Current Approaches to Clarification

The research literature on clarification in conversational table AI is dominated by two established approaches, always-answer and clarification question generation, plus a smaller line of exploratory work on uncertainty-aware response presentation.

**Always-answer** systems commit to the most likely interpretation of an ambiguous input and produce an answer without flagging the ambiguity. This is the default behaviour of most CTabQA and CText2SQL systems. It performs well on unambiguous inputs and on inputs where one interpretation is strongly dominant, but degrades when two or more interpretations are about equally likely, because it commits to one reading without signalling its uncertainty and fails whenever the user meant another.

**Clarification question generation (CQG)** systems aim to detect ambiguity and respond by generating a natural language question that prompts the user to resolve it. Deng et al. (2022) introduce PACIFIC, a conversational table QA dataset that incorporates clarification interactions within multi-turn reasoning. Yu et al. (2019a) present CoSQL, which includes clarification-like exchanges as part of a conversational Text-to-SQL framework. Elgohary et al. (2021) propose NL-EDIT, which addresses a related problem by enabling users to correct misgenerated SQL queries through natural language feedback, requiring the system to produce targeted clarifications that guide error resolution.

The central challenge of CQG is deciding *when* to ask. A system that asks for clarification on every mildly ambiguous input becomes tedious; a system that asks too rarely makes systematic errors. Deng et al. (2022) report a persistent gap to human performance on the clarification decision, suggesting that reliably estimating intent uncertainty remains challenging for current models. The broader survey by Deng et al. (2023) places this challenge in the context of proactive dialogue systems more generally, covering not only clarification but also target-guided dialogue (in which the system steers the conversation toward a pre-specified goal) and non-collaborative settings.

**Uncertainty-aware response presentation** is better viewed here as an early proposal than as a mature third paradigm. In this framing, the system produces multiple candidate answers (or SQL queries) corresponding to different interpretations, ranks them by estimated probability, and presents the top-ranked answer alongside an indication of its confidence. ABG-CoQA (Guo et al., 2021b) (an arXiv preprint; we are not aware of a widely used reproduction in the table-centric setting) studies ambiguity in conversational QA and argues that presenting multiple candidate answers with associated confidence scores can be preferable to committing to a single interpretation or always asking a clarification question, because it lets the user resolve ambiguity directly.

### 8.1.3   Cross-Category Synthesis

Taken together, the literature shows three shared weaknesses in clarification. First, explicit evaluation of clarification behaviour is rare outside PACIFIC and CoSQL, so the field still lacks a stable way to compare systems on when they ask, when they answer, and how well they handle ambiguity. Second, the threshold for asking versus answering is inconsistent across datasets and system designs, which makes clarification performance difficult to interpret even when it is reported. Third, the user's response to a clarification question is seldom modelled as a structured update to dialogue state; instead, it is often treated as just another utterance, which leaves systems vulnerable to repeating the same ambiguity or propagating the wrong assumption forward.

## 8.2 Dialogue Context Tracking

Every multiturn system must solve the problem of maintaining an accurate and useful representation of the conversational context accumulated across prior turns. Without this representation, the system cannot resolve coreferences, interpret context-dependent questions, or avoid repeating information already established in the dialogue. The form that this representation takes differs substantially across the five categories, but the underlying requirement is the same.

### 8.2.1 What Context Tracking Means in Each Category

In **CTabQA**, context tracking means maintaining a record of which entities, values, and columns have been discussed, what numerical results have been computed, and what the user appears to be trying to understand. The cTBLS system (Sundar & Heck, 2023) represents this through the conversation history and the retrieved table cells; ConvFinQA (Chen et al., 2022) represents it through a chain of numerical reasoning steps that carry intermediate results forward.

In **CText2SQL**, context tracking takes the specific form of SQL context: the current SQL query encodes which tables are selected, which conditions are active, which aggregations are in scope, and which columns are being returned. Systems such as IST-SQL (Wang et al., 2021) maintain an explicit interaction state that records this SQL context and updates it at each turn, while RASAT (Qi et al., 2022) encodes it implicitly through coreference relation types in the attention mechanism. HIE-SQL (Zheng et al., 2022) uses a history-enhanced encoder that directly attends over prior question and SQL pairs to resolve cross-turn dependencies.

In **Interactive Table Manipulation**, context tracking must follow not only what was said but also what was done. The table state changes with every executed operation, and subsequent instructions must be interpreted relative to the current table, not the original one. This mutable-state tracking is more demanding than tracking dialogue over a static table because the context the system must maintain includes the full history of table modifications, not just the history of utterances.

In **Agentic Table Systems**, context tracking expands further to include the outputs of all tool calls, the results of intermediate code executions, and the current state of all data artifacts that the agent has produced or modified during the session. ChatDB (Hu et al., 2023) externalises this context to a database, making it queryable and persistent across an arbitrarily long session.

### 8.2.2 What Makes Table Context Tracking Different

Context tracking over tables is not dialogue tracking with a table attached. Three properties make it its own problem.

**Referring by position, not by name.** In text dialogue, a follow-up usually refers to a named entity that appeared as a string in an earlier turn, so standard coreference tools can resolve it. In a table, the referent is often a cell fixed by its row and column, such as "that figure" meaning the value at *Q3 2023* and *Revenue*. The same number can sit in many cells, and the one the user means is set by position, not by wording. So the system has to track row and column coordinates, not just mentioned strings, and off-the-shelf coreference does not transfer.

**Carrying numbers across turns.** A table dialogue often computes a value in one turn and reuses it later without restating it, as in ConvFinQA. The state therefore has to hold intermediate numerical results, not only which entities were discussed. Systems that keep only the text of prior turns lose these values; the ones that work store them as named variables in generated code, which is why code generation has become the common way to handle numerical chaining.

**A state the user can change.** In querying and translating, the table is fixed, so the state is just the conversation. In manipulation and agentic settings, each action edits the table, so the state includes the current contents of an artifact the system itself keeps changing. Tracking this means recording what was done, not only what was said, and a wrong edit corrupts the state for every later turn.

These three properties are what the read-only versus mutable and numerical-chaining axes of Table 1 measure. They are also why the analogy to dialogue state tracking, which we draw next, is useful but incomplete.

### 8.2.3 The Dialogue State Tracking Analogy

The challenge of context tracking in conversational table AI is structurally analogous to dialogue state tracking (DST) in task-oriented dialogue systems, a well-studied problem in the spoken dialogue and conversational AI literature (Wu et al., 2020; Hosseini-Asl et al., 2020). In standard DST, the state is a set of slot-value pairs that represent the user's current goal (for example, in a hotel booking dialogue, the slots might be location, check-in date, price range, and room type). The state is updated at each turn as the user provides more information or changes their preferences.

In conversational table AI, the analogous state is richer and less structured. Rather than a flat set of slot-value pairs, the state includes the current SQL query (in CText2SQL), the history of retrieved cells and computed values (in CTabQA), the current table state (in Interactive Manipulation), and the current workflow execution state (in Agentic Systems). This richness is both an opportunity and a challenge: the richer state enables more complex interactions, but it also makes state maintenance harder and makes errors more difficult to detect and correct.

Despite this structural parallel, the DST and conversational table AI literatures have developed largely in parallel without substantial cross-fertilisation. Within CText2SQL specifically, this analogy also helps organise the model families surveyed in Section 5: SQL-editing methods approximate state updates, rewriting methods externalise context recovery, and IST-SQL (Wang et al., 2021) comes closest to an explicit state model. Question rewriting (Vakulenko et al., 2021), which converts a context-dependent question into a self-contained one by explicitly incorporating the relevant context, therefore acts as a concrete bridge between the two traditions. Integrating the principled state maintenance techniques developed for task-oriented DST with the richer state representations required for table interactions is a promising and underexplored research direction.

### 8.2.4 Cross-Category Synthesis

Across categories, context tracking remains brittle for three related reasons. First, most systems still rely on raw dialogue history rather than a structured, queryable summary of what remains in scope, what has already been computed, and which assumptions have been confirmed. Second, performance degrades with dialogue length wherever it has been measured (Chen et al., 2022), which suggests that current representations do not scale gracefully as state accumulates. Third, error recovery is largely absent: once a mistaken assumption enters the dialogue state, later turns usually inherit it rather than challenge, repair, or replace it.

## 8.3 Evaluation and Benchmarks

Evaluation is among the most fragmented aspects of conversational table AI. The five categories use different metrics, different benchmark formats, and different notions of what constitutes a correct response. This fragmentation makes it hard to compare progress across categories, to identify which capabilities are improving fastest, or to determine whether a new method represents a genuine advance across the field or only an improvement on a narrow benchmark.

### 8.3.1 The Benchmark Landscape

Table 2 provides a structured view of the benchmark landscape in two panels. Panel A collects the core multiturn and interactive benchmarks that anchor the survey taxonomy. Panel B isolates adjacent single-turn predecessors and background datasets that matter historically, but should not be overread as a directly comparable benchmark family with Panel A.

Table 3 deliberately separates *reporting conditions* from *performance values*. The point is not to rank benchmarks or to suggest a single cross-category leaderboard. It should not be used to infer that one category is stronger or more mature than another from one headline number. It is to show where public

Table 2: Benchmark landscape across the five taxonomy categories, presented in two panels to separate the core survey benchmarks from adjacent background predecessors. Category abbreviations: F = Foundations background, CQ = CTabQA, CS = CText2SQL, M = Interactive Manipulation, A = Agentic. Multiturn indicates context-dependent turns. Hybrid indicates table plus prose evidence. Primary metric abbreviations are defined in Section 8.3.2. Size is reported as number of dialogues or question sequences where available, and as number of individual QA pairs otherwise.

| Dataset | Cat. | Year | Multiturn | Domain | Hybrid | Size | Primary Metric |
|---|---|---|---|---|---|---|---|
| **Panel A. Core multiturn and interactive benchmarks in the survey taxonomy** | | | | | | | |
| WikiTableQ (Pasupat & Liang, 2015) | F | 2015 | No | Wikipedia | No | 22K pairs | FM Acc. |
| SQA (Iyyer et al., 2017) | CQ | 2017 | Yes | Wikipedia | No | 6,066 seq. | EM |
| SParC (Yu et al., 2019b) | CS | 2019 | Yes | 138 domains | No | 4,298 seq. | QEM, IEM |
| CoSQL (Yu et al., 2019a) | CS | 2019 | Yes | 138 domains | No | 3,007 dial. | EX, CQG, DA |
| HybriDialogue (Nakamura et al., 2022) | CQ | 2022 | Yes | Wikipedia | Yes | 4.8k dial. | ROUGE, EM |
| PACIFIC (Deng et al., 2022) | CQ | 2022 | Yes | Finance | Yes | 2,757 dial. | EM, CQG F1 |
| ConvFinQA (Chen et al., 2022) | CQ | 2022 | Yes | Finance | Yes | 3,892 dial. | Exec. acc. |
| MMCoQA (Li et al., 2022) | CQ | 2022 | Yes | General | Yes | 1K+ dial. | EM, F1 |
| SheetRM (Chen et al., 2025b) | M | 2024 | Yes | Office | No | varies | SubPass@1 |
| SpreadsheetBench (Ma et al., 2024) | M | 2024 | Yes | Office | No | varies | Pass@1 |
| InfiAgent-DABench (Hu et al., 2024) | A | 2024 | No | General | No | varies | Acc. by Qs. |
| DA-Code (Huang et al., 2024b) | A | 2024 | No | General | No | varies | Total acc. |
| iTBLS (Sundar et al., 2025) | CQ | 2025 | Yes | Scientific | No | 4K+ dial. | EM, BERTScore |
| **Panel B. Background single-turn or predecessor benchmarks** | | | | | | | |
| *These rows are included for lineage and context only. They are not intended to define a single directly comparable benchmark family with Panel A.* | | | | | | | |
| TAT-QA (Zhu et al., 2021) | CQ | 2021 | No | Finance | Yes | 16K pairs | EM, F1 |
| FinQA (Chen et al., 2021b) | CQ | 2021 | No | Finance | Yes | 8,281 pairs | Exec. acc. |
| BIRD (Li et al., 2023b) | CS | 2023 | No | 95 databases (37 domains) | No | 12K pairs | EX |

evaluation artifacts differ in split visibility, metric family, and task definition, because those differences are precisely what make score-level comparisons fragile across categories.

This separation is deliberate. Table 2 summarises the benchmark landscape itself. Table 3 records the public reporting conditions under which scores are exposed. Once the task definition, split, and public artifact differ, a single cross-category "best number" can imply comparability that the underlying benchmarks do not actually support.

Table 3: Public reporting conditions for selected core benchmarks. This is intentionally *not* a leaderboard table and must not be read as evidence of cross-category superiority. It records how results are usually reported, where they are exposed publicly, and what makes direct score comparisons difficult across categories.

| Benchmark | Cat. | Common public metric(s) | Typical public artifact | Split visibility | Main comparability caveat |
|---|---|---|---|---|---|
| WikiTableQuestions | F | Fuzzy Match accuracy | Paper and public benchmark reporting | Test | Important historical predecessor, but not a conversational benchmark |
| SQA | CQ | EM | Paper tables | Test | Decomposed sequential QA over Wikipedia tables; no explicit clarification task |
| HybriDialogue | CQ | ROUGE, EM | Paper tables | Test | Hybrid retrieval and generation setting; oracle-style retrieval variants also appear in the literature |
| PACIFIC | CQ | EM, clarification-question metrics | Paper tables | Test | Mixes answer quality with clarification quality within one benchmark |
| ConvFinQA | CQ | Execution accuracy | Paper tables | Dev and test reporting both appear in later work | Program-generation setting over hybrid financial documents, not plain span extraction |
| SParC | CS | QM, IM, execution-family metrics | Hidden-test leaderboard and paper tables | Dev and hidden test | Different papers emphasize different metric families, so exact-match and execution results should not be collapsed into one scalar |
| CoSQL | CS | EX, question / interaction metrics, dialogue-act and clarification metrics | Hidden-test leaderboard and paper tables | Dev and hidden test | Multitask benchmark; SQL generation, answer quality, and dialogue behavior are reported separately |
| SheetRM | M | Pass@1, Sub-Pass@1 | Paper tables | Test | Measures long action sequences over spreadsheets rather than open-ended free-form dialogue quality |
| SpreadsheetBench | M | Pass@1 | Public leaderboard and paper | Public leaderboard | Execution correctness over workbook states is central, which is not directly comparable to QA or SQL metrics |
| InfiAgent-DABench | A | Aggregate task accuracy | Project page and paper | Public benchmark reporting | Aggregates heterogeneous agentic tasks rather than a single dialogue objective |
| DA-Code | A | Aggregate task accuracy | Paper tables | Paper-reported splits | Mixes data-wrangling, machine-learning, and EDA subtasks in one evaluation bundle |

Several observations emerge from this reporting-context view. The benchmark landscape is dominated by the finance and Wikipedia domains. Office-focused benchmarks exist, but they typically evaluate scripted multi-step tasks rather than open-ended, user-driven dialogue with clarification and context repair. Agentic benchmarks are usually single-turn at the level of evaluation instances even though the systems themselves are multistep. A recent exception worth noting is BIRD-INTERACT (Huo et al., 2025), which scores text-to-SQL through dynamic interaction: each task starts from an ambiguous request and is graded over the model's exchanges with a user simulator, including when it should ask rather than answer. It is one of the few benchmarks that puts clarification and multiturn interaction at the centre of the evaluation rather than at the edge, which is exactly the gap noted in Section 8.1. Across categories, the reported metrics remain largely incomparable, which is precisely why a transparent statement of reporting conditions is more defensible than a single mixed leaderboard.

Table 4: Evaluation metrics that recur across the benchmark landscape, with their primary use cases and known limitations.

| Metric | Used In | What It Measures | Key Limitation |
|---|---|---|---|
| FM Acc. (Fuzzy Match) | Foundations (WTQ) | Approximate string match between prediction and gold answer under token/substring normalization | Can over-credit partially correct spans; sensitive to normalization heuristics and answer formatting |
| EM (Exact Match) | CTabQA | Whether predicted answer exactly matches the gold answer after normalization | Penalises paraphrases/synonyms; brittle to formatting and minor wording differences |
| EX (Execution Accuracy) | CText2SQL | Whether predicted SQL yields the same result set as the gold query when executed | Can be fooled by spurious queries that coincidentally return the same result; does not assess reasoning fidelity |
| Exec. acc. (program) | CTabQA (programmatic / tool-use QA) | Whether generated program/code (or intermediate tool outputs) executes to the correct final answer | Execution can be correct for the wrong reasons; depends on environment/tooling assumptions and error handling |
| ROUGE-1 | CTabQA (generative responses) | Unigram overlap between generated response and reference response | Weak proxy for factual correctness; can reward fluent but incorrect answers and punish valid paraphrases |
| F1 | CTabQA | Token-level overlap between prediction and gold answer (harmonic mean of precision/recall) | Still surface-form dependent; partial credit may not correspond to semantic correctness |
| SubPass@1 | Interactive Manipulation | Fraction of sub-tasks completed correctly on the first attempt | Sub-task definitions are benchmark-specific; not directly comparable across different manipulation suites |
| Pass@1 | Interactive Manipulation | Whether the full task completes correctly on the first attempt | Binary and coarse; does not distinguish near-misses from failures or measure efficiency/interaction cost |
| Accuracy by Questions | Agentic | Fraction of questions/tasks solved correctly (question-level accuracy) | Does not evaluate tool-call efficiency, reasoning trace quality, or robustness; can hide per-category failure modes |
| Total acc. (aggregate) | Agentic | Aggregate accuracy over multiple agent task types (e.g., DW/ML/EDA) under a benchmark's scoring protocol | Aggregation weights and task mix are benchmark-defined; overall score can mask weaknesses on critical subtasks |

### 8.3.2 Evaluation Metrics and Their Limitations

Table 4 summarises the primary evaluation metrics that recur across the benchmark landscape and reporting conditions above, together with their scope and known limitations.

The limitations enumerated in Table 4 point to a common underlying problem: every current metric measures a proxy for what we actually care about. What we care about is whether the system correctly understood what the user wanted, correctly identified the relevant data, and correctly produced a response that satisfies the user's goal. We are not aware of a metric that measures all three of these components together, or that measures them across a full multiturn dialogue without first breaking it into individual turns.

Each limitation in Table 4 has a concrete failure behind it, not just an abstract worry. Exact match marks a correct free-form answer wrong when it is worded differently, which is the reason FeTaQA (Nan et al., 2022) dropped span exact match in favour of generation metrics. Execution accuracy on a single database can credit a spurious query that returns the right rows by coincidence; distilled test-suite databases were introduced to catch exactly this case (Zhong et al., 2020). Surface metrics also miss structural understanding: models that score well on standard table benchmarks still fail on tasks that probe table layout and structure (Sui et al., 2024; Gu et al., 2025). The pattern is consistent, so the limitations column should be read as a list of known failure modes rather than hypothetical ones.

### 8.3.3 Fragmentation and Its Consequences

The evaluation fragmentation documented above has three practical consequences for the field.

First, it makes it difficult to identify genuine progress. A paper that reports a 3% improvement in QEM on SParC is not obviously comparable to a paper that reports a 5% improvement in Pass@1 on Spreadsheet-Bench. Both may represent equivalent amounts of scientific progress, or very different amounts. Without a shared metric, there is no way to know.

Second, it creates incentives to optimise for benchmark metrics at the expense of capabilities that are not measured. Systems trained to maximise execution accuracy on CoSQL, for example, have limited incentive to improve their clarification behaviour, because such behaviour is not directly or explicitly rewarded by the evaluation metrics, even though it is a genuine component of the task. The survey by Sui et al. (2024) systematically evaluates LLM capabilities on structured table data and demonstrates that models which score well on standard benchmarks often fail on structural reasoning tasks that require understanding the organisation and layout of a table rather than just its content. StrucText-Eval (Gu et al., 2025) similarly shows that current LLMs struggle with reasoning over structure-rich inputs, even when they perform strongly on content-oriented tasks.

Third, it prevents cumulative scientific progress. A new CTabQA system that achieves the best published result on HybriDialogue cannot be compared to a new CText2SQL system that achieves the best result on SParC, because the two tasks, the two datasets, and the two metrics are all different. Building a shared evaluation framework that spans at least a subset of the five categories is one of the most valuable infrastructure investments the conversational table AI community could make.

### 8.3.4 A Note on Multilinguality

The benchmark landscape surveyed here is almost entirely English. The clearest exception is in CText2SQL. CHASE (Guo et al., 2021a) is a large Chinese dataset for cross-database, context-dependent text-to-SQL. Outside text-to-SQL, multilingual conversational table AI is still thin, and we are not aware of a non-English, multiturn benchmark for table QA.

One source of friction is specific to this setting and deserves attention. The user often speaks one language while the database schema is named in English. Follow-up questions arrive in the user's language and lean on pronouns and ellipsis to point back to earlier turns. To answer them, the system has to do two things at once: resolve the reference in the user's language, then link it to an English column or table name that was never spoken. An entity introduced in an earlier native-language turn must be carried forward and matched to a schema element in a different language. Current systems are not built for this, and we are not aware of a benchmark that isolates it. Building multilingual CText2SQL and CTabQA datasets, especially for low-resource languages, is a useful and practical contribution.

### 8.3.5 Towards Better Evaluation

Three directions for improving evaluation are identifiable from the current literature.

**Process evaluation alongside outcome evaluation.** Current metrics measure the final output of a system but not the process by which it was produced. A system that generates a correct answer by accidentally retrieving the right cell has no more claim to having understood the question than one that retrieved

the wrong cell and guessed. Evaluating the evidence cited by a system, the intermediate reasoning steps it produced, and the confidence it expressed alongside its answer would provide a much richer picture of system capability than final-answer accuracy alone. The Table Meets LLM benchmark (Sui et al., 2024) and TabPedia (Zhao et al., 2024) represent steps toward richer evaluation of table understanding capabilities, though neither is specifically conversational.

**Dialogue-level metrics.** All current metrics evaluate individual turns in isolation. A metric that measures the coherence of a full dialogue, the degree to which the system's responses build on each other correctly across turns, the rate at which context errors propagate, and the quality of clarification exchanges would be more informative than any per-turn metric. Interaction-level exact match (IEM) is a crude step in this direction, but its binary all-or-nothing design makes it too strict to be informative in practice.

**Cross-category benchmarks.** We are not aware of a benchmark that tests a system on tasks spanning multiple categories, for example a task that requires answering a question (CTabQA), then editing the table based on that answer (Interactive Manipulation), and then generating a SQL query to verify the edit (CText2SQL). Such a benchmark would test the integrated capability that end-to-end conversational table AI requires, and its absence is a major structural gap in the current evaluation landscape.

# 9 Open Problems and Future Directions

Section 8 synthesised the recurring limitations that appear across CTabQA, CText2SQL, interactive manipulation, and agentic table systems. This section does not restate those issues category by category. Instead, it condenses them into six research problems whose resolution would most improve the field as a whole. These six problems absorb the recurring themes discussed earlier and therefore serve as the paper's single forward-looking agenda.

## 9.1 Unified Evaluation and Benchmark Design

A central structural gap in the field is still the absence of a shared evaluation framework. As Section 8.3 showed, categories reuse familiar metrics such as EM, execution accuracy, or Pass@1, but they instantiate them under different task definitions, annotation protocols, and test distributions, so scores are rarely comparable across categories in a defensible way. A useful next step is not one more category-specific leaderboard, but a benchmark suite that evaluates a common set of systems on tasks drawn from multiple categories under a shared protocol. That suite should pair outcome scoring with process annotations: which cells were used, which intermediate operations were performed, which clarification exchanges occurred, and how confidence changed across the dialogue. Without that infrastructure, the field will keep improving within silos without being able to show integrated progress. A convincing contribution here would deliver a benchmark suite, an annotation protocol, and a reporting template that multiple categories can all adopt without task-specific reinterpretation.

A related meta-research problem is to quantify the field's fragmentation directly rather than only describing it qualitatively. A practical starting point would be a bibliometric audit over the surveyed corpus: assign each paper to one primary category, construct a within- versus cross-category citation matrix, and report how citation flow changes over time. Such an analysis would not replace unified technical evaluation, but it would make cross-community isolation itself measurable and would test one of this survey's motivating observations with explicit evidence.

## 9.2 Dialogue State, Long-Horizon Coherence, and Error Recovery

Performance degrades with dialogue length wherever it has been measured, whether in ConvFinQA-style numerical dialogue, conversational SQL, or long operation sequences in manipulation benchmarks. The shared cause is that most systems still treat prior turns as raw history rather than as a compact, structured state capturing what remains in scope, what has already been computed, and which assumptions have been confirmed. This creates a second problem as well: once an early turn is wrong, later turns usually inherit that error rather than detect and repair it. A major advance would therefore be a dialogue-state representation

that supports retrieval, revision, and recovery, rather than merely longer context windows. Progress should be visible as flatter performance curves over turn depth and explicit recovery from injected earlier-turn errors.

### 9.3 Clarification and Ambiguity Management

Most current systems still guess when a user's request is ambiguous. Only a small subset of table-grounded benchmarks, most notably PACIFIC and CoSQL, evaluate clarification explicitly, and even there the field lacks a general account of when a system should ask, what it should ask, and how the answer should constrain subsequent reasoning. Treating clarification as a first-class capability requires three linked components: an ambiguity detector, a minimal and targeted clarification generator, and a state update mechanism that integrates the user's response into the ongoing dialogue. Until those three pieces are studied together, clarification will remain a special case rather than a core conversational competence. A useful benchmark here would score not just final accuracy, but whether the system asked only when needed, asked the right question, and updated its internal state correctly after the reply.

### 9.4 Numerical, Multi-Table, and Cross-Source Reasoning

Real analytical work rarely involves one table and one step. Users chain numerical results across turns, join multiple tables, compare documents from different sources, and shift attention between evidence types without restating every dependency explicitly. Current benchmarks usually isolate only one slice of that behaviour: ConvFinQA and PACIFIC stress numerical chaining, Spider 2.0 raises single-turn multi-table difficulty, and spreadsheet or agentic benchmarks cover only fragments of cross-source reasoning. A high-leverage research direction is therefore to model intermediate quantities, table provenance, and join structure as persistent conversational state rather than temporary by-products of a single turn. That would move the field closer to the actual structure of data analysis instead of its single-table approximation. A concrete success criterion would be sustained performance on multi-table, multi-document dialogues in which intermediate quantities must be reused several turns later.

### 9.5 Real-World Robustness, Hybrid Documents, and Domain Transfer

The benchmark distributions surveyed in this paper remain cleaner, flatter, and more English-centric than the environments in which these systems are meant to operate. Real spreadsheets contain merged cells, irregular layouts, hidden formulas, and embedded charts; real reports combine tables with prose and figures; real deployment settings require domain transfer into finance, medicine, science, law, and low-resource languages. These are often treated as separate problems, but they express the same underlying weakness: current systems are over-adapted to narrow benchmark conventions. The field needs benchmarks and models that preserve real structural messiness while remaining safe to release, for example through anonymisation pipelines, synthetic-yet-realistic table generation, and multilingual or domain-controlled evaluation protocols. A convincing step forward would show that a model trained under these conditions degrades gracefully across domains, layouts, and languages rather than collapsing outside its home benchmark.

### 9.6 Trustworthy Deployment: Interpretability, Privacy, and Safe Action

A deployable conversational table system must be auditable, privacy-aware, and safe to use on mutable data. That means more than producing a correct final answer: the system should expose the evidence cells it relied on, the operations it performed, the prior-turn assumptions it used, and enough provenance for a user to verify the result. It also means respecting access control, limiting schema leakage, and preventing destructive actions from being executed without confirmation, rollback, or human oversight. These concerns span answer generation, SQL execution, spreadsheet manipulation, and agentic workflows alike. If the field solves them only after benchmark accuracy plateaus, it will delay the transition from research demos to trustworthy analytical systems. Progress here should therefore be measured with deployment-facing criteria such as provenance exposure, rollback support, permission awareness, and user-verifiable action traces, not only end-task accuracy.

## 10 Conclusion

What this survey makes visible is that conversational AI over structured data is not a loose collection of niche tasks, but an emerging interaction stack. Foundations methods make tables legible to language models; CTabQA and CText2SQL study how intent and context are carried across turns; interactive manipulation systems add mutable state and execution; agentic systems add multistep planning and orchestration. Seen together, these areas reveal a single broader problem: building systems that can work with structured information the way people actually do, iteratively, contextually, and with the ability to ask, act, verify, and recover. That unifying view also clarifies why progress has felt uneven. Individual categories have advanced on their own benchmarked subproblems, but the capabilities needed for real deployment do not respect category boundaries. Clarification, long-horizon context tracking, numerical faithfulness, evidence grounding, robustness to messy and hybrid tables, and evaluation under realistic user goals all cut across the taxonomy. The main bottleneck is therefore not only model quality within each category, but the absence of shared evaluation and system designs that integrate these capabilities coherently. The next stage of the field should move from isolated task optimization toward integrated conversational data systems that can query, reason, manipulate, and orchestrate over structured information in transparent and user-aligned ways. We expect the most valuable future work to connect these capabilities rather than improve them in isolation.

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

## A  Terminology

The literature surveyed here uses inconsistent terminology for similar concepts. We adopt the following conventions throughout.

We use *turn* to mean a single exchange in a conversation: one user utterance and one system response. We use *dialogue* and *conversation* interchangeably for a sequence of turns. We use *session* when we wish to emphasize the full interactive episode, including any accumulated system state.

We use *utterance* for a single natural language input from the user, and *query* when that utterance is directed at a table with the intent to retrieve or compute information. We distinguish *query* in this sense, meaning a natural language expression of an information need, from *SQL query*, meaning a formal expression in Structured Query Language.

We use *grounding* for the process of connecting a natural language expression to specific elements of a table, namely identifying which cells, rows, columns, or values are relevant to a given utterance. Grounding is a prerequisite for answering questions, generating SQL, and producing table edits.

We use *context window* for the portion of the dialogue history that a system actively uses when processing the current turn. In early systems this was limited to the immediately preceding turn or a fixed window of recent turns; in LLM-based systems it may span the entire session subject to the model's maximum input length.

Table 5: Per-source search yield by query family (accessed 2026-06-14). DBLP is the primary, reproducible index (title/venue match); arXiv is a broad full-text indicator.

| Query family | DBLP | arXiv |
|---|---|---|
| conversational table question answering | 1 | 0 |
| conversational text-to-sql | 16 | 5 |
| spreadsheet agent | 14 | 1 |
| natural language interface to database | 108 | 44 |
| text-to-visualization | 105 | 481 |
| table agent | 88 | 4 |
| table foundation model | 7 | 1 |

## B  Review Protocol, Search Yield, and Audited Corpus

This appendix makes the review reproducible and the corpus auditable. It gives the search protocol and the exact queries (Section B.1), a dated per-source search yield (Section B.2), the inclusion and exclusion criteria (Section B.3), the citation-correction and deduplication record (Section B.4), and the full audited corpus list with every citation key bound to its verified source (Section B.5). The last item lets a reader confirm directly that the bibliography no longer contains key-to-paper mismatches.

### B.1  Search Protocol

We searched ACL Anthology, arXiv, DBLP, Semantic Scholar, OpenReview, and individual benchmark or project pages, and we expanded the seed works by backward and forward citation searching (snowballing). The keyword query families were:

1. "conversational table question answering"

2. "conversational text-to-sql"

3. "spreadsheet agent"

4. "natural language interface to database"

5. "text-to-visualization"

6. "table agent"

7. "table foundation model"

These were combined with the names of canonical datasets and systems anchoring each category. The corpus was frozen on March 27, 2026; later work is out of scope for this revision.

### B.2  Search Yield

Table 5 reports the number of records each query family returns from two reproducible indices, accessed 2026-06-14. DBLP matches query terms against publication titles and venues, so its counts are exact and reproducible; arXiv matches full text and does not strictly honour quoted phrases for hyphenated terms, so its counts run high (the 481 for "text-to-visualization" is a broad-match artifact). These are *identification* counts that indicate search breadth; most hits are out of scope, which is why the queries were combined with canonical names and relevance screening. Google Scholar is omitted because its counts are not reproducible across sessions. The counts are reproducible by re-running the released script (`search_yield.py`). These counts are post hoc reproducibility checks for two reproducible indices, not a PRISMA identification-to-inclusion flow. Because the original screening log was not preserved, we do not report PRISMA-style screened and excluded record counts.

## B.3   Inclusion and Exclusion

We kept a paper if it made a direct contribution to multiturn or interactive work over structured data, or if it provided background needed to explain one of the five categories. We excluded knowledge-graph systems, unstructured document dialogue without table grounding, table fact verification and table generation (which are single-turn and covered by prior surveys), and single-turn table tasks except where one was needed as a benchmark predecessor or historical background. Preprints and public artifacts were included when they introduced benchmarks, systems, or results not yet available in archival form, and the text labels their status.

Consistent with a structured narrative review, we make the queries, the search yield, the inclusion rules, the correction record, and the full audited corpus list available, which together let a reader reconstruct and check the evidence base.

## B.4   Citation Correction and Deduplication Record

During revision we audited every cited reference, checking that each citation key resolves to its intended paper, and we re-verified the CText2SQL and clarification clusters in full. As one cue for surfacing suspect entries we compared the author and year encoded in a key against the entry; this is only a heuristic, since citation keys are arbitrary labels and some encode an author or year that differs from the entry without being an error. We made the following corrections.

- Six keys named the correct work in the text but pointed to an unrelated paper in the bibliography. We corrected each: RASAT to Qi et al. (2022), IGSQL to Cai & Wan (2020), CoE-SQL to Zhang et al. (2024a), NL-EDIT to Elgohary et al. (2021), TOD-BERT to Wu et al. (2020), and SimpleTOD to Hosseini-Asl et al. (2020). (The first four are the mismatches identified by the reviewer; the last two were found by the same audit.)

- Two papers were entered twice. We merged the duplicates so each is cited once: "Table Meets LLM" (Sui et al., 2024) and SheetAgent (Chen et al., 2025b), and removed the redundant entries from the bibliography file.

- SheetRM is introduced within the SheetAgent paper rather than as a separate publication, so we cite Chen et al. (2025b) for it.

After these corrections the corpus contains 106 unique works with no duplicate entries, and we verified programmatically that no two cited keys share a DOI.

## B.5   Audited Corpus List

Table 6 lists all 106 cited works grouped by primary category, each shown with its citation key and the verified first author, year, title, and venue. This is the explicit audited corpus: a reader can confirm that every key resolves to the intended paper, which makes the citation audit directly verifiable. The citation key is an internal label and does not always encode the first author or year of the work it points to (for example, `liu2024alphasql` points to a 2025 paper whose first author is Li); what the table verifies is the binding from each key to the correct source.

Table 6: Audited corpus list: all 106 cited works, grouped by primary category. Each citation key is bound to its verified source.

| Citation key | First author (year) | Title | Venue |
|---|---|---|---|
| *Foundations* | | | |
| chen2024tablerag | Chen et al. (2024) | TableRAG: Million-Token Table Understanding with Language Models | NeurIPS |
| cheng2023binder | Cheng et al. (2023) | Binding Language Models in Symbolic Languages | ICLR |

*Table 6 continued*

| Citation key | First author (year) | Title | Venue |
|---|---|---|---|
| devlin2019bert | Devlin et al. (2019) | BERT: Pre-training of Deep Bidirectional Transformers for Language Understanding | NAACL |
| herzig2020tapas | Herzig et al. (2020) | TaPas: Weakly Supervised Table Parsing via Pre-training | ACL |
| jiang2022omnitab | Jiang et al. (2022) | OmniTab: Pretraining with Natural and Synthetic Data for Few-shot Table-based Question Answering | NAACL |
| liu2022tapex | Liu et al. (2022) | TAPEX: Table Pre-training via Learning a Neural SQL Executor | ICLR |
| su2024tablegpt2 | Su et al. (2024) | TableGPT2: A Large Multimodal Model with Tabular Data Integration | arXiv |
| wang2024chainoftable | Wang et al. (2024) | Chain-of-Table: Evolving Tables in the Reasoning Chain for Table Understanding | ICLR |
| xie2022unifiedskg | Xie et al. (2022) | UnifiedSKG: Unifying and Multi-Tasking Structured Knowledge Grounding with Text-to-Text Language Models | EMNLP |
| yin2020tabert | Yin et al. (2020) | TaBERT: Pretraining for Joint Understanding of Textual and Tabular Data | ACL |

**CTabQA**

| Citation key | First author (year) | Title | Venue |
|---|---|---|---|
| chen2020hybridqa | Chen et al. (2020) | HybridQA: A Dataset of Multi-Hop Question Answering over Tabular and Textual Data | COLING |
| chen2021finqa | Chen et al. (2021) | FinQA: A Dataset of Numerical Reasoning over Financial Data | EMNLP |
| chen2021ottqa | Chen et al. (2021) | Open Question Answering over Tables and Text | ICLR |
| chen2022convfinqa | Chen et al. (2022) | ConvFinQA: Exploring the Chain of Numerical Reasoning in Conversational Finance Question Answering | EMNLP |
| choi2018quac | Choi et al. (2018) | QuAC: Question Answering in Context | EMNLP |
| clark2016deep | Clark et al. (2016) | Deep Reinforcement Learning for Mention-Ranking Coreference Models | EMNLP |
| deng2022pacific | Deng et al. (2022) | PACIFIC: Towards Proactive Conversational Question Answering over Tabular and Textual Data in Finance | EMNLP |
| iyyer2017search | Iyyer et al. (2017) | Search-based Neural Structured Learning for Sequential Question Answering | ACL |
| jin2022medqa | Jin et al. (2020) | What Disease does this Patient Have? A Large-scale Open Domain Question Answering Dataset from Medical Exams | arXiv |
| khatuya2025finder | Khatuya et al. (2025) | Program of Thoughts for Financial Reasoning: Leveraging Dynamic In-Context Examples and Generative Retrieval | EMNLP |
| li2022mmcoqa | Li et al. (2022) | MMCoQA: Conversational Question Answering over Text, Tables, and Images | ACL |

*Table 6 continued*

| Citation key | First author (year) | Title | Venue |
|---|---|---|---|
| nakamura2022hybridialogue | Nakamura et al. (2022) | HybriDialogue: An Information-Seeking Dialogue Dataset Grounded on Tabular and Textual Data | COLING |
| nan2022fetaqa | Nan et al. (2022) | FeTaQA: Free-form Table Question Answering | TACL |
| pasupat2015compositional | Pasupat et al. (2015) | Compositional Semantic Parsing on Semi-Structured Tables | ACL |
| reddy2019coqa | Reddy et al. (2019) | CoQA: A Conversational Question Answering Challenge | TACL |
| shi2024ehragent | Shi et al. (2024) | EHRAgent: Code Empowers Large Language Models for Few-shot Complex Tabular Reasoning on Electronic Health Records | EMNLP |
| sui2024table | Sui et al. (2024) | Table Meets LLM: Can Large Language Models Understand Structured Table Data? A Benchmark and Empirical Study | WSDM |
| sundar2023ctbls | Sundar et al. (2023) | cTBLS: Augmenting Large Language Models with Conversational Tables | Proceedings of the . . . |
| sundar2024cpapers | Sundar et al. (2024) | cPAPERS: A Dataset of Situated and Multimodal Interactive Conversations in Scientific Papers | NeurIPS |
| sundar2025itbls | Sundar et al. (2025) | iTBLS: A Dataset of Interactive Conversations Over Tabular Information | Proceedings of the . . . |
| zhang2020bertscore | Zhang et al. (2020) | BERTScore: Evaluating Text Generation with BERT | ICLR |
| chen2021multihiertt | Zhao et al. (2022) | MultiHiertt: Numerical Reasoning over Multi Hierarchical Tabular and Textual Data | ACL |
| zhu2021tatqa | Zhu et al. (2021) | TAT-QA: A Question Answering Benchmark on a Hybrid of Tabular and Textual Content in Finance | ACL |
| ***CText2SQL*** | | | |
| cai2020igsql | Cai et al. (2020) | IGSQL: Database Schema Interaction Graph Based Neural Model for Context-Dependent Text-to-SQL Generation | EMNLP |
| cao2021lgesql | Cao et al. (2021) | LGESQL: Line Graph Enhanced Text-to-SQL Model with Mixed Local and Non-Local Relations | ACL |
| tracksql2025 | Chen et al. (2025) | Track-SQL: Enhancing Generative Language Models with Dual-Extractive Modules for Schema and Context Tracking in Multi-turn Text-to-SQL | COLING |
| dahl1994atis | Dahl et al. (1994) | Expanding the Scope of the ATIS Task: The ATIS-3 Corpus | HLT |
| gao2023dailsql | Gao et al. (2024) | Text-to-SQL Empowered by Large Language Models: A Benchmark Evaluation | VLDB |
| katsogiannis2023survey | Katsogiannis-Meimarakis et al. (2023) | A survey on deep learning approaches for text-to-SQL | VLDB |

*Table 6 continued*

| Citation key | First author (year) | Title | Venue |
|---|---|---|---|
| spider2024 | Lei et al. (2025) | Spider 2.0: Evaluating Language Models on Real-World Enterprise Text-to-SQL Workflows | ICLR |
| li2023bird | Li et al. (2023) | Can LLM Already Serve as A Database Interface? A BIg Bench for Large-Scale Database Grounded Text-to-SQLs | NeurIPS |
| liu2024alphasql | Li et al. (2025) | Alpha-SQL: Zero-Shot Text-to-SQL using Monte Carlo Tree Search | ICML |
| shi2022selfplay | Liu et al. (2022) | Augmenting Multi-Turn Text-to-SQL Datasets with Self-Play | COLING |
| liu2024nl2sql | Liu et al. (2024) | A Survey of NL2SQL with Large Language Models: Where are we, and where are we going? | arXiv |
| pourreza2024dinsql | Pourreza et al. (2023) | DIN-SQL: Decomposed In-Context Learning of Text-to-SQL with Self-Correction | NeurIPS |
| gao2024chasesql | Pourreza et al. (2025) | CHASE-SQL: Multi-Path Reasoning and Preference Optimized Candidate Selection in Text-to-SQL | ICLR |
| qi2022rasat | Qi et al. (2022) | RASAT: Integrating Relational Structures into Pretrained Seq2Seq Model for Text-to-SQL | EMNLP |
| scholak2021picard | Scholak et al. (2021) | PICARD: Parsing Incrementally for Constrained Auto-Regressive Decoding from Language Models | EMNLP |
| vakulenko2021question | Vakulenko et al. (2021) | Question Rewriting for Conversational Question Answering | WSDM |
| wang2020ratsql | Wang et al. (2020) | RAT-SQL: Relation-Aware Schema Encoding and Linking for Text-to-SQL Parsers | ACL |
| yu2021istsql | Wang et al. (2021) | Tracking Interaction States for Multi-Turn Text-to-SQL Semantic Parsing | AAAI |
| wang2024macsql | Wang et al. (2025) | MAC-SQL: A Multi-Agent Collaborative Framework for Text-to-SQL | COLING |
| yu2018spider | Yu et al. (2018) | Spider: A Large-Scale Human-Labeled Dataset for Complex and Cross-Domain Semantic Parsing and Text-to-SQL Task | EMNLP |
| yu2019sparc | Yu et al. (2019) | SParC: Cross-Domain Semantic Parsing in Context | ACL |
| yu2019cosql | Yu et al. (2019) | CoSQL: A Conversational Text-to-SQL Challenge Towards Cross-Domain Natural Language Interfaces to Databases | EMNLP |
| zhang2019editing | Zhang et al. (2019) | Editing-Based SQL Query Generation for Cross-Domain Context-Dependent Questions | EMNLP |
| xu2023actsql | Zhang et al. (2023) | ACT-SQL: In-Context Learning for Text-to-SQL with Automatically-Generated Chain-of-Thought | COLING |

*Table 6 continued*

| Citation key | First author (year) | Title | Venue |
|---|---|---|---|
| chen2024coesql | Zhang et al. (2024) | CoE-SQL: In-Context Learning for Multi-Turn Text-to-SQL with Chain-of-Editions | NAACL |
| wang2022hiesql | Zheng et al. (2022) | HIE-SQL: History Information Enhanced Network for Context-Dependent Text-to-SQL Semantic Parsing | COLING |
| *Manipulation* | | | |
| chen2023rigel | Chen et al. (2023) | Rigel: Transforming Tabular Data by Declarative Mapping | IEEE Trans. Vis.... |
| chen2024sheetagent | Chen et al. (2025) | SheetAgent: Towards a Generalist Agent for Spreadsheet Reasoning and Manipulation via Large Language Models | WWW |
| huang2024interactive | Huang et al. (2024) | Interactive Table Synthesis With Natural Language | IEEE Trans. Vis.... |
| li2024sheetcopilot | Li et al. (2023) | SheetCopilot: Bringing Software Productivity to the Next Level through Large Language Models | NeurIPS |
| ma2024spreadsheetbench | Ma et al. (2024) | SpreadsheetBench: Towards Challenging Real World Spreadsheet Manipulation | NeurIPS |
| narechania2021nl4dv | Narechania et al. (2021) | NL4DV: A Toolkit for Generating Analytic Specifications for Data Visualization from Natural Language Queries | IEEE Trans. Vis.... |
| setlur2016eviza | Setlur et al. (2016) | Eviza: A Natural Language Interface for Visual Analysis | UIST |
| srinivasan2017orko | Srinivasan et al. (2018) | Orko: Facilitating Multimodal Interaction for Visual Exploration and Analysis of Networks | IEEE Trans. Vis.... |
| tian2024spreadsheetllm | Tian et al. (2024) | SpreadsheetLLM: Encoding Spreadsheets for Large Language Models | arXiv |
| woods1973lunar | Woods et al. (1973) | Progress in natural language understanding: an application to lunar geology | American Federation... |
| zhang2024tablellm | Zhang et al. (2025) | TableLLM: Enabling Tabular Data Manipulation by LLMs in Real Office Usage Scenarios | COLING |
| huang2024sheetmind | Zhu et al. (2025) | SheetMind: An End-to-End LLM-Powered Multi-Agent Framework for Spreadsheet Automation | arXiv |
| *Agentic* | | | |
| hong2024metagpt | Hong et al. (2024) | MetaGPT: Meta Programming for A Multi-Agent Collaborative Framework | ICLR |
| hu2023chatdb | Hu et al. (2023) | ChatDB: Augmenting LLMs with Databases as Their Symbolic Memory | arXiv |
| hu2024infiagent | Hu et al. (2024) | InfiAgent-DABench: Evaluating Agents on Data Analysis Tasks | ICML |

*Table 6 continued*

| Citation key | First author (year) | Title | Venue |
|---|---|---|---|
| huang2024dacode | Huang et al. (2024) | DA-Code: Agent Data Science Code Generation Benchmark for Large Language Models | EMNLP |
| zhao2024talon | Jin et al. (2025) | TALON: A Multi-Agent Framework for Long-Table Exploration and Question Answering | EMNLP |
| lewis2020rag | Lewis et al. (2020) | Retrieval-Augmented Generation for Knowledge-Intensive NLP Tasks | NeurIPS |
| nam2025dsstar | Nam et al. (2025) | DS-STAR: Data Science Agent for Solving Diverse Tasks across Heterogeneous Formats and Open-Ended Queries | arXiv preprint |
| harmonia2025 | Santos et al. (2025) | Interactive Data Harmonization with LLM Agents: Opportunities and Challenges | arXiv preprint |
| schick2023toolformer | Schick et al. (2023) | Toolformer: Language Models Can Teach Themselves to Use Tools | NeurIPS |
| zhao2025tableagents | Tian et al. (2026) | Toward real-world Table Agents: capabilities, workflows, and design principles for LLM-based table intelligence | WWW (journal) |
| wei2022chain | Wei et al. (2022) | Chain-of-Thought Prompting Elicits Reasoning in Large Language Models | NeurIPS |
| yao2023react | Yao et al. (2023) | ReAct: Synergizing Reasoning and Acting in Language Models | ICLR |
| yang2024tablecritic | Yu et al. (2025) | Table-Critic: A Multi-Agent Framework for Collaborative Criticism and Refinement in Table Reasoning | ACL |
| zhang2024datacopilot | Zhang et al. (2023) | Data-Copilot: Bridging Billions of Data and Humans with Autonomous Workflow | arXiv |
| dbot2024 | Zhou et al. (2024) | D-Bot: Database Diagnosis System using Large Language Models | VLDB |
| wang2024autotqa | Zhu et al. (2024) | AutoTQA: Towards Autonomous Tabular Question Answering through Multi-Agent Large Language Models | VLDB |
| ***Background / Cross-cutting*** | | | |
| allen1999mixedinitiative | Allen et al. (1999) | Mixed-initiative interaction | arXiv preprint |
| chen2020tabfact | Chen et al. (2020) | TabFact: A Large-scale Dataset for Table-based Fact Verification | ICLR |
| codd1970relational | Codd et al. (1970) | A Relational Model of Data for Large Shared Data Banks | Commun. ACM |
| deng2023proactivedialogue | Deng et al. (2023) | A Survey on Proactive Dialogue Systems: Problems, Methods, and Prospects | AAAI |
| elgohary2021nl | Elgohary et al. (2021) | NL-EDIT: Correcting Semantic Parse Errors through Natural Language Interaction | NAACL |
| fang2024llmtabular | Fang et al. (2024) | Large Language Models (LLMs) on Tabular Data: Prediction, Generation, and Understanding - A Survey | Trans. Mach. Learn.... |

*Table 6 continued*

| Citation key | First author (year) | Title | Venue |
|---|---|---|---|
| fang2024structbench | Gu et al. (2025) | StrucText-Eval: Evaluating Large Language Model's Reasoning Ability in Structure-Rich Text | ACL |
| guo2021abgcoqa | Guo et al. (2021) | Abg-CoQA: Clarifying Ambiguity in Conversational Question Answering | 3rd Conference on . . . |
| guo2021chase | Guo et al. (2021) | Chase: A Large-Scale and Pragmatic Chinese Dataset for Cross-Database Context-Dependent Text-to-SQL | ACL |
| hosseini2020simple | Hosseini-Asl et al. (2020) | A Simple Language Model for Task-Oriented Dialogue | NeurIPS |
| huo2025birdinteract | Huo et al. (2025) | BIRD-INTERACT: Re-imagining Text-to-SQL Evaluation for Large Language Models via Lens of Dynamic Interactions | arXiv |
| ji2021survey | Ji et al. (2022) | A Survey on Knowledge Graphs: Representation, Acquisition, and Applications | IEEE Trans. Neural . . . |
| page2021prisma | Page et al. (2021) | The PRISMA 2020 statement: an updated guideline for reporting systematic reviews | Syst. Rev. |
| scaffidi2005estimating | Scaffidi et al. (2005) | Estimating the Numbers of End Users and End User Programmers | 2005 IEEE Symposium . . . |
| wu2020tod | Wu et al. (2020) | TOD-BERT: Pre-trained Natural Language Understanding for Task-Oriented Dialogue | EMNLP |
| huang2023conversational | Zaib et al. (2021) | Conversational Question Answering: A Survey | arXiv preprint |
| zhang2024nlinterfaces | Zhang et al. (2024) | Natural Language Interfaces for Tabular Data Querying and Visualization: A Survey | IEEE Trans. Knowl. . . . |
| zha2023tabpedia | Zhao et al. (2024) | TabPedia: Towards Comprehensive Visual Table Understanding with Concept Synergy | NeurIPS |
| zhong2020testsuite | Zhong et al. (2020) | Semantic Evaluation for Text-to-SQL with Distilled Test Suites | EMNLP |

# C  Extended Background

## C.1  What Is a Table?

For the purposes of this survey, a *table* is any data structure that organizes information into rows and columns, where each column represents a named attribute and each row represents a single entity, observation, or record. This definition is intentionally broad. It covers relational database tables, where rows are called tuples and columns are typed attributes governed by a schema; spreadsheet grids, where cells may contain numerical values, text, or formulas; financial report tables, which appear embedded in PDF documents and often combine numerical values with textual footnotes; and result tables in scientific publications.

Not all structures that look like tables share the same properties. Table 7 summarizes the main types and three properties that matter for language-based interaction. **Schema regularity** is whether the column names and data types are fixed and declared in advance: relational tables are regular, financial tables often

Table 7: Types of tabular data encountered across the five categories of this survey. ✓ indicates the property is typically present; ✗ indicates it is typically absent.

| Table Type | Regular Schema | Structurally Complex | Hybrid Content |
|---|---|---|---|
| Relational database table | ✓ | ✗ | ✗ |
| Spreadsheet (office) | ✗ | ✓ | ✗ |
| Wikipedia table | ✗ | ✓ | ✓ |
| Financial report table | ✗ | ✓ | ✓ |
| Scientific result table | ✗ | ✓ | ✓ |
| CSV / flat file | ✓ | ✗ | ✗ |

are not. **Structural complexity** is whether the layout is flat or contains nested headers, merged cells, or subtotals: Wikipedia and financial tables are often complex, CSV and relational tables are flat. **Hybrid content** is whether the table stands alone or sits alongside prose that qualifies it: financial reports and scientific papers are hybrid, so a question about a footnote may point to text beside the table rather than a cell inside it.

These differences have direct consequences. A system that works well on flat relational tables may fail on financial report tables with merged column headers, and tracking dialogue context over a clean database schema is qualitatively different from tracking it over a scientific table with abbreviated condition names.

### C.2 What Makes Tabular Data Different from Text?

Much of the machinery developed for conversational text understanding does not transfer directly to tables. Four reasons stand out.

**Position encodes meaning.** The meaning of a cell depends on its row and column. The value 142.3 means nothing without its column header ("Revenue in millions") and row identifier ("Q3 2023"). Language models pretrained on running text are not natively aware of this structure, so it must be added through pretraining objectives, as in TaPas (Herzig et al., 2020) and TaBERT (Yin et al., 2020), or through serialization that flattens the table while preserving position.

**Numerical reasoning is often required.** Many table questions need arithmetic. Language models are not calculators, and early systems that pattern-matched over serialized text did poorly on arithmetic-heavy benchmarks such as FinQA (Chen et al., 2021b) and ConvFinQA (Chen et al., 2022). Recent approaches offload arithmetic to an interpreter by generating code or SQL.

**Scale varies enormously.** A Wikipedia table has a few dozen cells; an enterprise database has thousands of tables with millions of rows; a financial-modeling spreadsheet spans hundreds of sheets with cross-sheet formulas. Techniques tuned for small tables do not necessarily scale.

**Tables occur in context.** Many real tables are embedded in documents whose prose explains or qualifies them, so strong performance on isolated-table benchmarks does not predict performance on hybrid documents.

## D Reference Tables

The tables below give the per-item detail behind the comparisons in the main text. Each is cited from the section where its claims are discussed.

Table 8: Positioning this survey relative to adjacent surveys. "Multiturn central" asks whether context-dependent dialogue is a primary organizing principle rather than a peripheral topic. "Manipulation / agentic" asks whether spreadsheet actions, tool use, or orchestration over structured data are part of the survey's core scope.

| Survey | Primary scope | Multiturn central | Manipulation / agentic | What it does not center |
|---|---|---|---|---|
| Fang et al. (Fang et al., 2024) | LLMs on tabular data broadly, including prediction, synthesis, question answering, and table understanding | No | No | Multiturn interaction over structured data as the main organizing axis |
| Liu et al. (Liu et al., 2024) | NL2SQL / Text-to-SQL methods, data, evaluation, and error analysis | Partial | No | Cross-community treatment of QA, manipulation, and agents |
| Zhang et al. (Zhang et al., 2024b) | Natural language interfaces for tabular querying and visualization | Partial | Partial | A taxonomy centered on dialogue state, clarification, and cross-category interaction |
| Katsogiannis and Koutrika (Katsogiannis-Meimarakis & Koutrika, 2023) | Deep learning Text-to-SQL systems and benchmarks | No | No | Conversational interaction beyond SQL generation |
| Zaib et al. (Zaib et al., 2021) | Conversational QA broadly as a multiturn dialogue problem | Yes | No | Structured-table interaction as the main object of study |
| This survey | Conversational AI over structured data across five categories | Yes | Yes | – |

Table 9: Summary of the five taxonomy categories. The interaction mode column uses the typology defined in Section 2.2. Representative papers are listed to orient the reader before the detailed sections that follow.

| Category | Mode | Primary Task | Representative Works | Section |
|---|---|---|---|---|
| Foundations | enabling substrate[†] | Table encoding, serialization, pretraining | TaPas (Herzig et al., 2020); TaBERT (Yin et al., 2020); TAPEX (Liu et al., 2022b); TableGPT2 (Su et al., 2024) | 3.2 |
| Conversational Table QA (CTabQA) | Querying | Multiturn natural language question answering over tables or hybrid documents | SQA (Iyyer et al., 2017); HybriDialogue (Nakamura et al., 2022); PACIFIC (Deng et al., 2022); ConvFinQA (Chen et al., 2022) | 4 |
| Conversational Text-to-SQL (CText2SQL) | Translating | Context-dependent natural language to SQL generation across dialogue turns | SParC (Yu et al., 2019b); CoSQL (Yu et al., 2019a); RASAT (Qi et al., 2022); CoE-SQL (Zhang et al., 2024a) | 5 |
| Interactive Table Manipulation | Manipulating (+ exploration) | Natural language driven modification, exploration, and generation of table content | SheetCopilot (Li et al., 2023a); SheetAgent (Chen et al., 2025b); Eviza (Setlur et al., 2016) | 6 |
| Agentic Table Systems | Orchestrating | Multistep autonomous planning and execution over structured data | Data-Copilot (Zhang et al., 2023b); InfiAgent-DABench (Hu et al., 2024); AutoTQA (Zhu et al., 2024) | 7 |

[†] Foundations is not itself one of the four primary interaction modes defined in Section 2.2; it is included separately because representation, serialization, and table pretraining methods enable all four modes.

Table 10: Representative methods in the Foundations category. "Training style" is used instead of "pretraining objective" because some methods are primarily inference-time frameworks rather than newly pretrained models. Input/access format: Flat = sequence serialization; Struct = structure-aware encoding; RAG = retrieval-augmented; ICL = in-context learning.

| Method | Year | Training style | Input/access | Scale | Key contribution |
|---|---|---|---|---|---|
| TAPAS (Herzig et al., 2020) | 2020 | MLM + weak supervision | Flat + table embeddings | BERT-Large | Extends BERT to table–text inputs with row and column embeddings for cell selection and aggregation |
| TaBERT (Yin et al., 2020) | 2020 | MLM + MCP + CVR | Struct | BERT | Joint text–table pretraining with vertical attention over rows and column-aware encoding |
| TAPEX (Liu et al., 2022b) | 2022 | SQL-exec pretraining | Flat | BART-Large | Trains a seq2seq model to mimic SQL execution for table reasoning |
| OmniTab (Jiang et al., 2022) | 2022 | Natural + synthetic pretraining | Flat | BART-Large | Combines web table–text data and SQL-derived synthetic QA in one table QA framework |
| UnifiedSKG (Xie et al., 2022) | 2022 | Unified text-to-text multitask learning | Struct | T5-3B | Frames 21 structured knowledge grounding tasks in a single text-to-text setup |
| Binder (Cheng et al., 2023) | 2023 | Training-free neural-symbolic ICL | Code + table | Codex | Generates executable programs with API calls to support symbolic reasoning over tables |
| Chain-of-Table (Wang et al., 2024) | 2024 | ICL with table operations | Struct + op history | LLM-based | Uses intermediate table transformations as the reasoning chain instead of free-form CoT |
| TableRAG (Chen et al., 2024) | 2024 | RAG + program-aided reasoning | RAG | LLM-based | Scales table QA with query expansion plus schema and cell retrieval |
| TableGPT2 (Su et al., 2024) | 2024 | Encoder alignment + instruct tuning | Struct + encoder | 7B / 72B | Couples a semantic table encoder with an LLM for stronger table-centric reasoning |

Table 11: Comparison of CTabQA-related datasets. Size is reported using each dataset's native counting unit(s). Evidence: T = table, T+X = table plus text, M = multimodal (e.g., text + tables + images, or paper components such as tables, figures, and equations). Eval. lists the primary evaluation families used in the original paper. [†] TAT-QA and FinQA are single-turn predecessors included because PACIFIC and ConvFinQA are derived from them, respectively.

| Dataset | Year | Size | Domain | Evidence | Eval. |
|---|---|---|---|---|---|
| SQA (Iyyer et al., 2017) | 2017 | 6,066 seq. (17,553 QAs) | Wikipedia | T | Overall acc., seq. acc. |
| HybriDialogue (Nakamura et al., 2022) | 2022 | 4,844 conv. | Wikipedia | T+X | MRR/MAP; Sacre-BLEU/BERTScore |
| PACIFIC (Deng et al., 2022) | 2022 | 2,757 conv. (19,008 turns) | Finance | T+X | CNP: P/R/F1; CQG: ROUGE-2/EM/F1; CQA/PCQA: EM/F1 |
| ConvFinQA (Chen et al., 2022) | 2022 | 3,892 conv. (14,115 Qs) | Finance | T+X | ExeAcc, ProgAcc |
| MMCoQA (Li et al., 2022) | 2022 | 1,179 conv. (5,753 QAs) | General | M | Recall/NDCG; F1/EM |
| cPAPERS (Sundar et al., 2024) | 2024 | 5,030 QA pairs | Scientific | M | ROUGE, METEOR, BERTScore, BLEU |
| iTBLS (Sundar et al., 2025) | 2025 | 4,000 tables | Scientific | T | EM |
| TAT-QA (Zhu et al., 2021)[†] | 2021 | 16,552 Qs (2,757 ctx) | Finance | T+X | EM, F1 |
| FinQA (Chen et al., 2021b)[†] | 2021 | 8,281 examples | Finance | T+X | ExeAcc, ProgAcc |

Table 12: Comparison of datasets relevant to CText2SQL, ordered by year. Context indicates whether the correct output depends on prior turns or iterative workflow context. EX denotes execution accuracy. EM denotes Spider-style exact set match. QM and IM denote question-level and interaction-level exact set match, respectively. SR denotes success rate. DA denotes dialogue-act prediction accuracy. LCR denotes logic correctness rate.

| Dataset | Year | Size | Context | DBs | Collection | Primary Metrics |
|---|---|---|---|---|---|---|
| ATIS (Dahl et al., 1994) | 1994 | 8,297 train utt.; 3,211 test utt. | Partial | 1 | Multi-site spoken dialogue | Task-specific ATIS evaluation |
| Spider (Yu et al., 2018) | 2018 | 10,181 Q; 5,693 SQL | No | 200 | Student annotators | EM, EX |
| SParC (Yu et al., 2019b) | 2019 | 4,298 interactions; 12k+ Q | Yes | 200 | Controlled user interactions | QM, IM |
| CoSQL (Yu et al., 2019a) | 2019 | 3,007 dialogues; 30k+ turns; 10k+ SQL | Yes | 200 | Wizard-of-Oz | QM, IM, BLEU, LCR, DA acc. |
| BIRD (Li et al., 2023b) | 2023 | 12,751 Q–SQL pairs | No | 95 | Expert-curated | EX |
| Spider 2.0 (Lei et al., 2025) | 2024 | 632 workflow problems | Multi-step | 213 | Expert-curated enterprise workflows | SR (Spider 2.0); EX (lite/snow) |

Table 13: Representative CText2SQL systems organized by architectural generation, reported with a *consistent exact-match metric family.* QM = question match and IM = interaction match. Because the cited papers do not all report on the same split, we explicitly list the evaluation split used in each source. For Track-SQL, results depend on the backbone; we report the stronger DeepSeek-7B variant reported in the cited paper. CoE-SQL reports official test-set execution-based question and interaction scores separately in the original paper, so we discuss those figures in prose below instead of mixing them directly into this exact-match table. MAC-SQL is included for completeness but does not report SParC/CoSQL results in the cited paper.

| System | Generation | Core Mechanism | SParC QM | SParC IM | CoSQL QM | CoSQL IM | Split |
|---|---|---|---|---|---|---|---|
| Edit-SQL (Zhang et al., 2019) | SQL editing | Copies prior SQL and edits it for the current turn | 47.9 | 25.3 | 40.8 | 13.7 | test |
| IGSQL (Cai & Wan, 2020) | Graph encoder | Schema interaction graph across turns | 51.2 | 29.5 | 42.5 | 15.0 | test |
| IST-SQL (Wang et al., 2021) | State tracking | Explicit schema-state and SQL-state tracking | 47.6 | 29.9 | 41.8 | 15.2 | SParC dev, CoSQL test |
| HIE-SQL (Zheng et al., 2022) | History-enhanced encoder | History-aware bimodal encoder with schema linking | 64.6 | 42.9 | 53.9 | 24.6 | test |
| RASAT (Qi et al., 2022) | Relation-aware | T5 with relation-aware self-attention and coreference relations | 67.7 | 49.1 | 58.8 | 27.0 | test |
| ACT-SQL (Zhang et al., 2023a) | LLM prompting | In-context learning with automatically generated chain-of-thought | 51.0 | 24.4 | 46.0 | 13.3 | dev |
| CoE-SQL (Zhang et al., 2024a) | LLM prompting | Chain-of-editions prompting from prior SQL | 56.0 | 36.5 | 52.4 | 23.9 | dev |
| Track-SQL (Chen et al., 2025a) | Dual-extractive | Separate schema extractor and context extractor modules | 65.17 | 46.44 | 58.19 | 28.67 | dev |
| MAC-SQL (Wang et al., 2025) | Multi-agent | Decomposition, tool use, and refinement by collaborating agents | N/A | N/A | N/A | N/A | N/A |

Table 14: Representative systems in the Interactive Table Manipulation category and closely related chart-oriented interfaces. Operation types covered: E = entry/value editing, M = structural or worksheet management, F = formatting, W = formula writing, T = table transformation (including pivot-style restructuring), C = chart generation or chart interaction. "Benchmark / Eval." denotes the primary benchmark or evaluation setting reported by the paper. For user-study systems, the final column summarises the main reported quantitative finding rather than Pass@1. N/A indicates that a directly comparable benchmark score was not reported.

| System | Year | Operations | Benchmark / Eval. | Backbone | Main Result |
|---|---|---|---|---|---|
| SheetCopilot (Li et al., 2023a) | 2023 | E, M, F, W, T, C | SCB | GPT-3.5 / GPT-4 | 55.0 Pass@1[†] |
| SheetAgent (Chen et al., 2025b) | 2025 | E, M, F, T, C | SCB, SheetRM | GPT-3.5 | 61.1 Pass@1 (SCB); 44.8 Pass@1 / 77.0 SubPass@1 (SheetRM) |
| SheetMind (Zhu et al., 2025) | 2025 | E, M, T | Self-curated tasks | Gemini | 80% single-step; ~70% multi-step success |
| TableLLM (Zhang et al., 2025) | 2025 | E, M, T, C | Modified WikiSQL, filtered Spider, self-created op. benchmark | Llama 3.1-8B | 89.6 (WikiSQL mod.); 77.8 (op. bench.) |
| Rigel (Chen et al., 2023) | 2023 | T | User study | Rule-based | N/A |
| NL2Rigel (Huang et al., 2024a) | 2024 | T | Comparative user study | GPT-3.5 | Same completion rate as Rigel (86/96), with lower time cost |
| Eviza (Setlur et al., 2016) | 2016 | C | User study | Rule-based | N/A |
| NL4DV (Narechania et al., 2021) | 2021 | C | Toolkit / case studies | Rule-based | N/A |
| Orko (Srinivasan & Stasko, 2018) | 2018 | C | User study | Rule-based | N/A |

[†] Reported on a subset of SCB with GPT-4. On the full SCB benchmark, SheetCopilot reports 44.3 Pass@1 with GPT-3.5.

Table 15: Representative agentic artifacts surveyed in this section. Artifact kind: Sys = system/framework, Proto = prototype, Bmk = benchmark. Agent setup: S = single-agent, M = multi-agent, − = not applicable. Setting: DB = relational/database-native tasks, TR = general table reasoning / long-table QA, EHR = multi-tabular electronic health records, HT = heterogeneous data-analysis sources. Tool use indicates whether the artifact invokes external tools such as code execution, SQL engines, or retrieval/external operations. Eval.: B = automated benchmark, H = human evaluation, D = scenario demonstration.

| Artifact | Year | Kind | Agents | Setting | Tools | Eval. | Primary Contribution |
|---|---|---|---|---|---|---|---|
| Data-Copilot (Zhang et al., 2023b) | 2023 | Sys | S | HT | Yes | B | Code-centric autonomous data analysis with offline interface discovery and workflow deployment |
| AutoTQA (Zhu et al., 2024) | 2024 | Sys | M | DB | Yes | B | Multi-agent tabular QA across multiple tables and database systems |
| Table-Critic (Yu et al., 2025) | 2025 | Sys | M | TR | No | B | Collaborative criticism and iterative refinement for table reasoning |
| TALON (Jin et al., 2025) | 2025 | Sys | M | TR | Yes | B | Long-table exploration and QA with planning, tool, and critic agents |
| DA-Code (Huang et al., 2024b) | 2024 | Bmk | − | HT | Yes | B | Benchmark for agent-based data-science code generation |
| InfiAgent-DABench (Hu et al., 2024) | 2024 | Bmk | − | HT | Yes | B | Benchmark for end-to-end data-analysis agents with execution feedback |
| Harmonia (Santos et al., 2025) | 2025 | Proto | S | HT | Yes | D | Interactive data harmonization with LLM reasoning and integration primitives |
| D-Bot (Zhou et al., 2024) | 2024 | Sys | M | DB | Yes | B,H | Multi-agent database diagnosis with tool use and cross-review |
| ChatDB (Hu et al., 2023) | 2023 | Sys | S | DB | Yes | B | SQL databases as symbolic memory for LLM reasoning |
| EHRAgent (Shi et al., 2024) | 2024 | Sys | S | EHR | Yes | B | Code-driven agent for few-shot complex multi-tabular EHR reasoning |

