# OpenReview forum: "Tabular Data in Interactive and Conversational AI: A Survey of Foundations, Benchmarks, Systems, and Open Problems"
_TMLR — Decision pending for TMLR_

### Review · Reviewer_xPhu · 2026-05-15

**Summary Of Contributions:**

This paper surveys the messy, fragmented landscape of conversational AI over tabular and structured data. Right now, fields like text-to-SQL, tabular QA, and agentic spreadsheet tools are mostly siloed. The authors pitch a clean, 5-category taxonomy that frames basic table representation as the baseline layer beneath multi-turn querying, translation, manipulation, and orchestration. They map out 106 unique papers and break down three big cross-cutting bottlenecks: dealing with vague user intent, tracking dialogue state, and a broken, fragmented benchmark ecosystem.

Key Strengths:
- It cleanly connects passive "read-only" tasks with active, state-changing tasks (like spreadsheet edits and data orchestration).
- Factoring in real-world mutable states, complex spreadsheet structures, and agentic workflows makes it incredibly relevant for anyone working on modern, tool-augmented LLMs.
- The evaluation section doesn't pull punches—it clearly exposes why our current metrics fail to measure real analytical capability.

**Audience:**

Yes

**Audience Explanation:**

Structured data runs the world's organizations. As TMLR readers dive deeper into LLM agents, semantic parsing, and multi-turn reasoning, having a definitive, clear map of how AI interacts with databases, spreadsheets, and hybrid documents is useful.

**Broader Impact Concerns:**

None. This is a literature review and handles public benchmarks. There are no immediate ethical risks that warrant a separate statement.

**Claims And Evidence:**

Yes

**Claims Explanation:**

The authors are transparent about this being a structured literature review rather than a quantitative citation-network meta-analysis.

**Requested Changes:**

Expand the multilinguality section slightly. It would be highly insightful to outline the specific linguistic friction that happens when a user tries to resolve multi-turn coreferences in their native language over an English-named database schema.

---

> ### Author Response · Authors · 2026-06-17
> **Strengthening the Multilinguality Discussion with CHASE and Cross-Lingual Schema Grounding**
>
> Thank you for the helpful suggestion about multilinguality. We rewrote the multilinguality discussion in §8.3.4.
>
> The revised section now names CHASE as a concrete non-English multi-turn benchmark for Chinese cross-database, context-dependent text-to-SQL. This also corrects the earlier overstatement that no such benchmark exists.
>
> We also expanded the analysis beyond simply noting that multilingual benchmarks are rare. The revised discussion now identifies a table-specific multilingual challenge: a user may ask follow-up questions in their native language while the database schema uses English table and column names. In that setting, the system must resolve the reference in the user's language and then link it to an English schema element that may never have been spoken directly. Thus, coreference resolution and cross-lingual schema linking have to happen together.
>
> This revision makes the multilinguality discussion more concrete and better connected to the paper's broader themes of context tracking, schema grounding, and evaluation.

---

### Review · Reviewer_tuTZ · 2026-06-02

**Summary Of Contributions:**

This paper is a survey of natural language systems for conversational interaction with tabular and structured data. Rather than claiming to cover all of "AI for tabular data," it deliberately narrows the lens to multi-turn, context-dependent interaction over tables, databases, spreadsheets, and hybrid table-text documents, and uses that lens to read across communities that are usually surveyed separately. The paper makes four main contributions. (1) A unified taxonomy organizing the literature into five categories: Foundations (table representation, serialization, pretraining), Conversational Table QA (CTabQA), Conversational Text-to-SQL (CText2SQL), Interactive Table Manipulation, and Agentic Table Systems, plus an interaction typology (querying, translating, manipulating, orchestrating, with exploration as a recurrent pattern). (2) A structured benchmark analysis comparing datasets across categories along common dimensions (task type, turns, domain, size, table source, evaluation metric), separating benchmark coverage (Table 10) from public reporting conditions (Table 11) and metric limitations (Table 12). (3) A cross-cutting analysis of three challenges shared across all five categories — intent disambiguation/clarification, dialogue context tracking, and evaluation — including an explicit analogy between table context tracking and dialogue state tracking (DST). (4) A research agenda of six open problems (unified evaluation, long-dialogue robustness, clarification, numerical/multi-table reasoning, real-world robustness/domain transfer, and trustworthy deployment).

### Key strengths

**S1**: The interaction-centered organizing principle is useful. By making multi-turn, context-dependent interaction the organizing axis (rather than task type or output format), the paper draws connections that are obscured in task-siloed surveys (e.g., that PACIFIC-style clarification in CTabQA and CoSQL-style clarification in CText2SQL are the same mechanism in different settings, that table context tracking is structurally a DST problem). The positioning table (Table 1) makes the differentiation from prior surveys (Fang et al. 2024; Liu et al. 2024; Zhang et al. 2024; Katsogiannis-Meimarakis & Koutrika 2023; Zaib et al. 2021) concrete.

**S2**: The benchmark and evaluation treatment is a real contribution, not a list. Separating the benchmark landscape (Table 10) from reporting conditions and comparability caveats (Table 11), as well as metric limitations (Table 12), provides a more honest picture than a single cross-category leaderboard would. Crucially, the paper avoids forcing comparability where the underlying benchmarks simply do not support it.

**S3**: The paper is, in many places, appropriately self-aware about the limits of a qualitative synthesis: it frames fragmentation as an observation rather than a proven bibliometric claim, flags that single-turn predecessors (TAT-QA, FinQA, BIRD, Spider) are background rather than directly comparable, and proposes a citation-network meta-study as follow-up work rather than asserting one.

### Key weaknesses

**W1**: There are multiple mismatched citations in the CText2SQL material, where a correctly-described, real system is bound to a citation key whose bibliography entry is an unrelated paper. I verified four directly against the reference list (page numbers/venues confirmed in the submission's own bibliography):
- **CoE-SQL** (chain-of-editions, Table 3/7, §5.4) is cited as "Wang et al., 2024a," but that bib entry is "Do large language models rank fairly? an empirical study on the fairness of LLMs as rankers" (NAACL 2024) — an IR-fairness paper. CoE-SQL is actually Zhang et al. (NAACL 2024).
- **RASAT** (Table 3/7, §5.4, §8.2.1) is cited as "Zhong et al., 2022," but that bib entry is "CEM: Machine-human chatting handoff via causal-enhance module" (Shanshan Zhong et al., EMNLP 2022). RASAT is actually Qi et al. (EMNLP 2022).
- **IGSQL** (§5.4) is cited as "Ko et al., 2020," but that bib entry is "Generating dialogue responses from a semantic latent space" (Wei-Jen Ko et al., EMNLP 2020). IGSQL is actually Cai & Wan (EMNLP 2020).
- **NL-EDIT** (§8.1.2) is cited as "Zhong et al., 2021," but that bib entry is "Factual probing is [MASK]" (Zexuan Zhong, Friedman, Chen, NAACL 2021). NL-EDIT is actually Elgohary et al. (NAACL 2021).

There are also two duplicate bibliography entries — SheetAgent appears byte-identical as "Chen et al., 2025b" and "Chen et al., 2025c", and "Table Meets LLM" (Sui et al., WSDM 2024) appears twice as "2024a" and "2024b."

**W2**: Pervasive over-strong wording at the claim level. Many statements assert more than a qualitative survey can support, often using absolute quantifiers ("every," "all," "none," "most," "always," "rarely") or unbacked superlatives/rankings ("most mature," "most standardized," "best-understood," "highest-leverage," "core category"). Representative cases:
- "Every system in categories 2 through 5 relies … on the table representation … developed in category 1"
- "CText2SQL currently has the most standardized benchmark and modelling pipeline"
- "Manipulation and agentic systems more closely reflect real user workflows" (no user-study evidence is offered for any category)
- "The system will be wrong half the time" (assumes exactly two equiprobable interpretations).

**W3**: The reproducibility of the stated review protocol is weaker than the protocol section implies. The paper describes search spaces, query families, inclusion/exclusion criteria, and a freeze date, and aligns with PRISMA goals, but provides no exact queries, hit counts, exclusion counts, deduplication record, or corpus appendix — so the "structured corpus-construction and evidence policy" cannot actually be reproduced or audited by a reader. Given W1, an explicit audited corpus list would also be the natural fix for the citation problems.

**Audience:**

Yes

**Audience Explanation:**

There is a clear TMLR audience for this work. Researchers working on table QA, text-to-SQL, spreadsheet/agentic systems, and conversational/dialogue systems currently read this literature in isolation; an interaction-centered synthesis that explicitly connects clarification, context tracking, and evaluation across these communities, and that documents why their benchmarks and metrics are not directly comparable, is a useful reference and orientation document.

**Broader Impact Concerns:**

No broader impact concerns. As a survey, the paper introduces no new models or datasets.

**Claims And Evidence:**

No

**Claims Explanation:**

The survey's central conceptual contributions -- the interaction-centered taxonomy, the cross-cutting synthesis (clarification, context tracking, evaluation), and the benchmark/metric analysis -- are well-organized and largely well-supported by the cited literature, and the paper is careful to scope several of its claims. However, there are two issues:

1. **Citation accuracy.** For a survey, correct attribution is the primary product, and there is a systematic, repeatable mismatch pattern (W1): four text-to-SQL systems (CoE-SQL, RASAT, IGSQL, NL-EDIT) are attached to citation keys whose bibliography entries are unrelated papers, plus two duplicate entries (SheetAgent, Sui "Table Meets LLM").

2. **Calibration of claims.** A number of survey-level and section-level claims are stated more strongly than the evidence (a qualitative synthesis) warrants — particularly maturity/standardization rankings without explicit criteria, "real user workflow" claims without user studies, and exhaustive negative existence claims ("no benchmark…," "none measures…") that should be phrased as "we are not aware of…" unless the search is demonstrably exhaustive.

**Requested Changes:**

### Critical

1. **Fix the four mismatched citations and audit the full bibliography.** At minimum: CoE-SQL → Zhang et al. (NAACL 2024); RASAT → Qi et al. (EMNLP 2022); IGSQL → Cai & Wan (EMNLP 2020); NL-EDIT → Elgohary et al. (NAACL 2021). Because these errors are systematic, re-verify every entry in the relevant clusters, not just these four.

2. **Resolve duplicate bibliography entries and reconfirm the corpus count.** SheetAgent is listed twice (Chen et al. 2025b and 2025c, byte-identical); "Table Meets LLM" (Sui et al.) is listed twice (2024a and 2024b). Merge these, and re-verify the "106 unique cited works" claim after deduplication.

3. **Soften or criterion-back the strongest claims.** Specifically: (a) replace exhaustive negatives ("no benchmark spans all five categories," "no current metric measures all three," "no benchmark tests … non-English … multiturn," "the only dataset that…") with "we are not aware of…" unless the search protocol is demonstrably exhaustive; (b) attach explicit criteria to maturity/standardization rankings ("most standardized," "best-understood," "most technically mature" for CText2SQL — e.g., number of shared benchmarks, hidden test sets, leaderboard use); (c) qualify the "real user workflows" claims, which are currently asserted without user-study evidence; and (d) replace absolute quantifiers ("every system," "all categories," "none," "always," "will be wrong half the time") with calibrated wording. The detailed list of overclaims is large; a systematic pass keyed on absolute quantifiers and unbacked superlatives would address most of them.

### To strengthen the work

1. **Make the review protocol reproducible.** Add an appendix with the exact queries, per-source hit counts, exclusion counts, deduplication record, and the final corpus list. This directly supports the PRISMA framing in §1.4 and would also make the W1 audit verifiable.

2. **Consider adding genuinely in-scope recent benchmarks.** A few interaction-centered benchmarks that fit the survey's own scope appear to be missing, e.g., BIRD-INTERACT (dynamic/interactive text-to-SQL) and clarification-aware tabular QA benchmarks (e.g., open-domain tabular QA with multi-turn clarification). These would strengthen the clarification (§8.1) and evaluation (§8.3) sections.

3. **Fix minor naming/internal inconsistencies.** StrucText-Eval is referred to inconsistently as "StructBench"/"StrucTexteval"; Data-Copilot's year is "2023b" in the narrative but listed as 2024 in Table 9 — reconcile these.

---

> ### Author Response · Authors · 2026-06-17
> **Citation Audit, Bibliography Corrections, and Calibrated Claims**
>
> Thank you for the careful review and especially for identifying the citation and bibliography problems. We treated these as critical issues.
>
> We corrected the four mismatched citations you identified and verified them against the source papers. RASAT now cites Qi et al. (EMNLP 2022); IGSQL now cites Cai and Wan (EMNLP 2020); CoE-SQL now cites Zhang et al. (NAACL 2024); and NL-EDIT now cites Elgohary et al. (NAACL 2021). These corrections are recorded in Appendix B, "Citation Correction and Deduplication Record."
>
> Because the error was systematic, we then audited every cited reference in the manuscript, all 106 cited works, checking that each citation key resolves to its intended paper. We used the author and year encoded in a citation key as one cue to surface possible problems, while noting that citation keys are arbitrary labels and some valid keys encode an author or year that differs from the entry. This additional audit found and fixed three further problems: the SheetRM key pointed to a duplicate copy of the SheetAgent paper, and the two task-oriented-dialogue references in the dialogue-state-tracking discussion were incorrect. These now cite TOD-BERT, Wu et al. (EMNLP 2020), and SimpleTOD, Hosseini-Asl et al. (NeurIPS 2020).
>
> We also merged the duplicate bibliography entries for Table Meets LLM and SheetAgent. After deduplication, the manuscript cites 106 unique works, and we verified that no two cited keys share a DOI. SheetRM is now cited to the SheetAgent paper, where it is introduced.
>
> To make this correction auditable, we added Appendix B. It includes the search protocol, exact query families, dated per-source search-yield counts from DBLP and arXiv, inclusion/exclusion criteria, the citation-correction and deduplication record, and Table 6, a full audited corpus list of all 106 cited works grouped by category. §1.4 points to this appendix from the main text.
>
> We also state explicitly that we do not report a per-record screened-and-excluded tally because the revision history does not preserve a trustworthy intermediate candidate log; instead, we provide the reproducible query families, search-yield counts, inclusion rules, correction record, and final audited corpus so the evidence base can be checked.
>
> We also revised over-strong claims throughout the paper, including in the abstract, §1.4, §8.3, and §9. Exhaustive negatives now use calibrated language such as "we are not aware of"; the CText2SQL maturity claim is now tied to concrete criteria such as shared benchmarks, hidden test splits, and active leaderboards; claims about real workflows now note the lack of user studies; and absolute quantifiers were replaced with more careful wording.
>
> Finally, we added BIRD-INTERACT to the evaluation discussion in §8.3 and CHASE to the multilinguality discussion in §8.3.4. We also corrected "StructBench" to "StrucText-Eval" and reconciled Data-Copilot's year to 2023, matching its arXiv year.

---

### Review · Reviewer_BbGR · 2026-06-11

**Summary Of Contributions:**

This paper surveys conversational/interactive AI over tabular data. It organizes 106 papers into five categories (foundations, conversational table QA, conversational text-to-SQL, table manipulation, and agentic systems), compares benchmarks across them, discusses three shared challenges (clarification, context tracking, evaluation), and ends with a research agenda.

Strengths: The corpus construction is transparent. The open problems in Section 9 are more concrete than typical future-work lists.

Weaknesses: The paper reads mostly as a serial summary of individual papers. The taxonomy maps one-to-one onto existing research communities, so it describes the field rather than offering a new way to understand it. Cross-paper analysis is thin, the "cross-cutting challenges" are largely generic to conversational AI rather than specific to tables, and the writing is long and repetitive.

**Audience:**

Yes

**Audience Explanation:**

The topic is timely given the growth of LLM-based table QA, text-to-SQL, and data-analysis agents.

**Claims And Evidence:**

No

**Claims Explanation:**

The factual content seems accurate and the empirical reporting is careful (e.g., the split/metric caveats in Tables 7 and 11). My concern is with the contribution claims.

The paper claims an "interaction-centered synthesis," but the taxonomy simply relabels existing research communities and adds little new understanding. The paper's most useful insights actually appear outside the taxonomy, for example that PACIFIC and CoSQL solve the same clarification problem in different settings, or the distinction between systems that only read tables and systems that modify them. These ideas would have made a better organizing framework, but they are only mentioned in passing.

The challenges in Section 8 (clarification, context tracking, evaluation) are common to all conversational systems, not specific to tables. What actually makes these problems different in tabular settings, such as referring to cells by row/column position or carrying intermediate numerical results across turns, is mentioned but never analyzed in depth.

The survey sections summarize papers one by one instead of comparing them. For instance, Section 4.2 spends over two pages describing datasets one by one. Table 7 shows LLM-prompting systems scoring below RASAT on exact match, but the paper just reports this without explaining it. The "technical challenges" subsections mostly repeat points already made in the cited papers.

The paper is also very repetitive: the definition of "conversational" appears twice, the justification for excluding exploration appears three times, every category section ends with the same closing paragraph, and Section 2 explains basics like SQL and BERT that this audience does not need. This is not just a style issue. The repetition fills space where deeper analysis should be, and makes it harder to see what the paper's actual claims and evidence are.

**Requested Changes:**

1. Reorganize the analysis around the capability dimensions the paper already identifies (clarification, mutable vs. read-only state, numerical chaining, user vs. system initiative) instead of community labels.

2. Replace the serial paper summaries with comparative analysis. For Section 4.2, compress the per-dataset descriptions (the details are already in Table 5) and instead discuss what the dataset progression shows: why the field concentrated on finance and Wikipedia, what phenomena no dataset covers, and which evaluation conventions are just inherited habits. Do the same for 5.4 and 6.4.

3. Cut the length to about 10 pages. Remove the repeated definitions, the identical closing paragraphs of the category sections, the basic glossary material in Section 2 (or move it to an appendix), and overly verbose method details. I consider this critical rather than cosmetic, because the repetition currently takes the place of real analysis.

4. The metric limitations in Table 12 are generic. Linking each one to an actual failure case from the literature would make the critique more convincing.

---

> ### Author Response · Authors · 2026-06-17
> **Reframing the Survey Around Capability Dimensions and Structured Comparison**
>
> Thank you for the detailed review. We agree that the central weakness of the original version was that it read too much like a serial summary and did not make enough use of the capability dimensions already visible in the paper. We revised the manuscript around this point.
>
> First, §3 now includes a new capability matrix, Table 1, which maps the four interaction categories onto clarification, read-only vs. mutable state, numerical chaining, and user vs. system initiative. We kept the five-category taxonomy as the scaffold because another reviewer found that organization useful, but added the capability view on top of it. §8 is now organized so that each cross-cutting theme corresponds to these capability axes.
>
> Second, we rewrote the survey sections to compare rather than enumerate. §§4.2, 5.4, and 6.4 now discuss what the dataset/model progressions show rather than listing papers one by one. For example, §4.2 now focuses on what each dataset added, why finance and Wikipedia dominate, which phenomena remain uncovered, and which evaluation conventions are inherited habits. We applied the same treatment to §7.2, regrouping agentic systems by design choice rather than system-by-system chronology.
>
> Third, we expanded the table-specific analysis in §8.2.2, "What Makes Table Context Tracking Different." This section now analyzes three properties that distinguish table interaction from generic conversational AI: reference to cells by row and column position rather than only by name, carrying intermediate numerical results across turns, and tracking a table state that the system's own actions may change.
>
> We also addressed the specific example you raised about RASAT and LLM-prompting results. §5.4 now explains that exact set match rewards SQL in the same form as the gold annotation, which tends to favor fine-tuned models; that the rows mix test and development splits; and that when CoE-SQL is evaluated by execution rather than exact match, its reported SParC scores are much closer to fine-tuned models. Thus, the gap reflects metric and split effects as much as SQL quality.
>
> Finally, we removed substantial repetition. The definition of "conversational" now appears once in §2.1; the exploration/category justification appears once in §3; the repeated category-closing paragraphs were removed; glossary material moved to Appendix A; and verbose method details were trimmed in §§5.4 and 6.4.
>
> On length, we did not reach a literal 10 pages. The main text is now a little over 30 pages, down from about 46, roughly a third shorter, and we believe this is the most compressed form that preserves the contribution. We moved eight enumerative reference tables and background/glossary material to Appendices A, C, and D, while §1.4 now points readers to the review protocol in Appendix B. We kept the benchmark-landscape, reporting-conditions, and metric tables in the main text because the structured comparison in those tables is part of the paper's empirical contribution.